# The Detection of *Wolbachia* in Tea Green Leafhopper (*Empoasca onukii* Matsuda) and Its Influence on the Host

Qiuqiu Zhang [1], Rongmeng Lan [1], Dezhong Ji [1], Yanni Tan [1], Xia Zhou [1], Xiaofeng Tan [2], Qiong Wu [2] and Linhong Jin [1,*]

[1] State Key Laboratory Breeding Base of Green Pesticide and Agricultural Bioengineering, Key Laboratory of Green Pesticide and Agricultural Bioengineering, Ministry of Education, Guizhou University, Huaxi District, Guiyang 550025, China; gs.zhangqq18@gzu.edu.cn (Q.Z.); gs.rmlan19@gzu.edu.cn (R.L.); gs.dzji19@gzu.edu.cn (D.J.); gs.yntan20@gzu.edu.cn (Y.T.); fcc.xzhou@gzu.edu.cn (X.Z.)

[2] Plant Protection and Quarantine Station of Guizhou Province, Guiyang 550001, China; tanxf@163.net.cn (X.T.); wuqiong2008@163.com (Q.W.)

* Correspondence: lhjin@gzu.edu.cn; Tel.: +86-186-8517-4719

**Abstract:** Tea green leafhopper (*Empoasca onukii* Matsuda) is a critical pest in tea production. *Wolbachia* has attracted much attention as a new direction of pest biological control for its ability of manipulating the hosts' reproductive biology. In this work, we focused on the detection of *Wolbachia* in tea green leafhopper and its effect on host reproduction and development. Polymerase chain reaction (PCR), real-time PCR, and fluorescence in situ hybridization (FISH) techniques were used to detect the distribution of *Wolbachia* in tea green leafhopper. *Wolbachia* infection levels were different in different organs of hosts in different insect stages. In addition, comparison between the infected populations and cured population (treated by tetracyclines) revealed that presence of *Wolbachia* apparently influenced the growth, life cycle, and other reproductive factors of tea green leafhopper, caused, for example, by cytoplasmic incompatibility (CI), thereby reducing number of offspring, shortening lifespan, and causing female-biased sex ratio. This research confirmed that the bacteria *Wolbachia* was of high incidence in tea leafhoppers and could significantly affect the hosts' reproductive development and evolution.

**Keywords:** tea green leafhopper; *Wolbachia*; reproductive development; cytoplasmic incompatibility

## 1. Introduction

The tea green leafhopper, (*Empoasca onukii* Matsuda) (Hemiptera: Cicadellidae), is the major pest affecting tea production in China [1]. The tea green leafhopper, because of its small size and high mobility, is usually unnoticeable until it proliferates in large numbers. It has hemimetabolous development and has 9 to 11 generations a year in the tea producing areas [2,3]. A tea green leafhopper pierces and sucks the leaf sap and causes tea seedlings to wither and become brown [4,5] and the female adults lay eggs in the young tea shoots, which ultimately leads to serious loss on the quality and yield of tea production [6–9]. The controlling method of this pest mainly relies on frequent use of pesticides, which might lead to pesticide residues in tea, resistance of insects, pollution of the environment, and destruction of natural enemies [4,10,11]. Since the diversity and distribution of symbiotic bacteria *Wolbachia* has been studied in many groups of arthropods including leafhoppers [12–14], it is expected to be an eco-friendly alternative management method to control tea leafhopper by inhibiting or antagonizing symbiotic bacteria and realize the green and sustainable development of tea production.

An endosymbiont lives within the body or cells of another insect forming an endosymbiosis and interacting with the host in a symbiotic or parasitic way [15–18]. Some examples of endosymbiosis are bacteria [17–19], microsporidia [20], and RNA viruses [21], which

can manipulate reproduction of the host. The well-studied endosymbiont *Wolbachia*, a genus of intracellular bacteria, can infect arthropod species, including a high proportion of insects [22]. Studies estimate that approximately 40% of terrestrial arthropod species [23] and more than 16% of insects in the world are potential hosts of *Wolbachia* [24]. Its interactions with its hosts are often complex, and in some cases have evolved to be mutualistic rather than parasitic [22]. *Wolbachia* is usually responsible for three types of reproductive manipulation of host, namely male killing, feminization, and cytoplasmic incompatibility (CI) [22,25–27]. Some *Wolbachia* strains have a negative effect on hosts by changing the sex ratio or resulting in a shorter insect life [22,26]. CI occurs when a *Wolbachia*-infected male mates with an uninfected female or when mating occurs between males and females infected with different *Wolbachia* strains, giving birth to progeny that die during very early embryogenesis and reducing the number of offspring [28,29]. So, *Wolbachia* is of considerable interest nowadays and has potential for use as a biocontrol agent due to its ubiquitous distribution and various evolutionary functions [30–34].

Antibiotics application could change or cure *Wolbachia* infection which in turn could influence the offspring production [35–39]. For example, antibiotic treatment cured *Wolbachia* infection in *Encarsia Formosa* and affected hosts' reproductive development [35]. These reports concluded that the symbiotic bacteria *Wolbachia* has impact on the reproductive development of host and can be cured or regulated by antibiotics.

However, information regarding *Wolbachia* infection and influence on tea leafhopper in tea agroecosystems has not been investigated, except in a survey about different symbiotic bacterial infections in tea garden insects [40]. Here, we revealed the *Wolbachia* infection status of tea green leafhoppers from different geographical tea gardens and variation in infection rates of *Wolbachia* in different insect stages and followed the potential influence on the development and reproduction of tea green leafhopper by using polymerase chain reaction (PCR)-based detection, fluorescence in situ hybridization (FISH), and tetracycline treatment comparison.

## 2. Materials and Methods

### 2.1. Insect Collection

Sample Collection: The adults of tea green leafhopper were collected from four counties in Guizhou province, at an average height of about 1000 m, at geographical coordinates of 106–107 degrees north(N) and 25–27 degrees east(E), including Pingtang (106° E–25° N, height 1487.7 m), Meitan (107° E–27° N, height 972.7 m), Jiu'an (106° E–26° N, height 1236.2 m), and Fenggang (107° E–27° N, height 809 m). Some of the samples were stored in a −20 °C refrigerator and directly used for gene detection of *Wolbachia*'s distribution.

In addition, tea green leafhopper's eggs that were adhered to branches were collected in Jiu'an and hatched in a climate-controlled room (26 ± 1 °C, 65 ± 5%) of relative humidity (RH), 16 h:8 h light(L)/dark(D). Fresh tea branches were provided to feed the insects. The tea branches were cultured with nutrient solution [41], which was replaced every 5 days to keep the tea branches fresh.

This insect rearing method was applied for tea green leafhopper and its offspring in the study of *Wolbachia's* effects on insect reproductive development.

### 2.2. Genome Sequencing

2.2.1. DNA Preparation

Genomic DNA from leafhoppers was extracted and examined for the presence of *Wolbachia* using *Wolbachia*-specific surface protein (*wsp*) primers [42]. A Biospin Insect Genomic DNA Extraction Kit (Biomars, Beijing, China) was used for total DNA extraction.

The whole body of tea leafhopper (five insects in a group) was ground using a sterile plastic pestle in a 1.5 mL centrifuge tube containing 200 μL ddH$_2$O, 20 μL proteinase K, and 200 μL Buffer gA. The homogenate was incubated at 56 °C for 1 h and transferred to a spin column and centrifuged at 12,000× *g* for 1 min. Extracted DNA in column was washed with 500 μL Buffer PW twice and then 55 μL TE buffer was applied to elute and obtain



the DNA solution, which was then stored at $-20\ ^{\circ}$C for future use. The concentration of elution was determined by a Nanodrop (IMPLEN). 1 µL of the solution was used as template DNA subjected to PCR-screen for examining *Wolbachia* infection.

### 2.2.2. PCR Amplification

The *wsp*1 (81F: TGGTCCAATAAGTGATGAAGAAAC; 691R: AAAAATTAAACGC-TACTCCA), *16S rRNA* (F: CGGGGGAAAATTTATTGCT; R: AGCTGTAATACAGAAAG-GAAATCGCCA) and *FtsZ* (F: ATYATGGARCATATAAARGATAG; R: TCRAGYAATG-GATTRGATA) [42,43] were used for amplification. The PCR reaction mixture contained 1.0 µL of genomic DNA template, 5 µL of 10× buffer, 4 µL of dNTP's (2.5 mM), 2.0 µL of each primer (10 nM of Primer F and Primer R), 0.5 µL of Pyrobest DNA Polymerase (5 U/µL). ddH$_2$O was added to the mixture to make a final 50 µL reaction solution. The PCR program was initially started by 32 cycles of amplification at 94 $^{\circ}$C for 30 s, 55 $^{\circ}$C for 30 s, and 72 $^{\circ}$C for 60 s, followed by one cycle of amplification at 94 $^{\circ}$C for 30 s, 52 $^{\circ}$C for 30 s, and 72 $^{\circ}$C for 60 s, and a final extension at 72 $^{\circ}$C for 60 min. PCR was performed using a 2720 Thermal Cycler (Applied Biosystems, USA). The PCR products were separated on a 1% (*w/v*) agarose gel stained with ethidium bromide (EB) and photographed by a gel imaging system.

### 2.3. Real-Time PCR

Symbiotic bacterial infection was detected on real-time PCR by using the *wsp*2 (F: TG-GAACCCGCTGTGAATGAT; R: GCACCATAAGAACCGAAATAACG). The PCR reaction mixture was prepared as follows formula: 5 µL of 2× TSINGKE' Master real-time PCR Mix-SYBR, 0.5 µL of Primer F and Primer R, 1 µL of DNA template, and proper ddH$_2$O added to achieve 10 µL final reaction solution. The reaction mixture was initially denatured at 95 $^{\circ}$C for 600 s; followed by 40 cycles of amplification at 95 $^{\circ}$C for 10 s, 60 $^{\circ}$C for 10 s, and 72 $^{\circ}$C for 10 s using a Light Cycler$^{\circledR}$ 96 (Roche, Basel, Switzerland).

### 2.4. Fluorescence In Situ Hybridization (FISH)

Hybridization nucleotide probe W2(5′-CTTCTGTGAGTACGTCATTATC-3′) in FISH was applied to the nymphs and adults of tea green leafhopper collected from Jiu'an. The probe was a 5′-labeled quasar 670 (excitation wavelength: 647 nm; emission wavelength: 670 nm) according to Jureemart Wangkeeree's method [44]. Fresh samples were collected and stored in acetone until further use. Nymphs and adults were immersed in phosphate-buffered saline (PBS; 137 mM NaCl, 2.7 Mm KCl, 10 mM Na$_2$KPO$_4$, and 2 mM KH$_2$PO$_4$), and cuticles were pricked with a needle in two or three places under a microscope to facilitate reagent infiltration. All samples were fixed overnight in Carnoy's solution (ethanol: chloroform: acetic acid, 6:3:1) at room temperature (28–35 $^{\circ}$C) in a shaker. The samples were decolorized by immersion and 6% (*v/v*) hydrogen peroxide in ethanol at room temperature (28–35 $^{\circ}$C) for 2–3 weeks to quench the autofluorescence of the insect tissues. After decolorization, the samples were soaked three times in absolute ethanol/PBST [1:1, *v/v* (PBST: PBS with 0.2% Tween-20)] solution for 10 min each time. Before hybridization, the samples were hydrated three times in hybridization buffer (20 mM Tris–HCl [pH 8], 0.9 M NaCl, 0.01% [*v/v*] SDS, and 30% [*v/v*] formamide) for 10 min each time.

For the hybridization reaction, buffers containing 100 nM probe were added to the samples, and the mixtures were incubated overnight at 46 $^{\circ}$C in the dark. Non-specifically bound probes were washed three times with PBS for 10 min each time. Samples were finally mounted on a slide with a lid using a slide imaging confocal laser scanning microscope at Abace Biology. Reactions without probes or RNase digestion before hybridization were used as negative controls [45].

### 2.5. Antibiotic (Tetracycline) Treatment on Wolbachia Hosted Tea Green Leafhopper

As reported [37], tetracycline can eliminate the symbiotic bacteria *Wolbachia* in insects. According to the requirements of breeding and experiment, we designed tetracycline

solutions with different concentrations (1 mg/mL, 5 mg/mL, and 10 mg/mL). Different concentrations of tetracycline solution were smeared on the tea tree twigs and leaves, which then were dried slowly. Subsequently, 20 newly hatched tea green leafhopper nymphs in each group were transferred into the fresh tea leaves in a container. Tetracycline was applied to the leaves once a day, and water served as control.

The tea green leafhoppers were treated with different concentrations of tetracycline and dynamically observed for 35 days every 5 days.

### 2.6. The Effect of Wolbachia on the Reproductive Development of the Tea Green Leafhopper

The naturally *Wolbachia*-infected populations and uninfected (tetracycline-treated populations) were compared for their longevity in different stages.

Newly emerged adults (two pairs) were randomly selected and raised in self-made vials, at 25 ± 1 °C, L/D = 16:8, and RH 65 ± 5%. Then, 15 replicates were set for each treatment. The fresh tea buds were provided and replaced every day. The eggs of tea green leafhopper attached to tea buds were collected and the number was recorded. Those eggs were then gathered in an artificial climate box and moisturized with wet filter paper until they hatched. After hatching, the newly emerged nymphs were reared with fresh tea buds and observed every 12 h for their reproductive parameters, such as the time span of each stage (spawning, egg, nymph, and adult stages), until their death.

### 2.7. Reproductive Regulation of Wolbachia in Tea Green Leafhopper

The populations (tea green leafhopper collected from Jiu'an) that were naturally infected with *Wolbachia* were marked as "W". "W−" represented tetracycline-cured populations, which were *Wolbachia*-free (uninfected). The mating design was set as follows: W−♀×W♂, W♀×W♂, W♀×W−♂, W−♀×W−♂(♀female, ♂male). One male and one female were matched as a crossing pair and mated in the same enclosed environment. The sex ratio of their offspring was recorded.

### 2.8. Data Analysis

*Wsp* gene concentration in total DNA was calculated following the formula: *wsp* gene copy numbers (copies/μL) = $6.02 \times 10^{23}$ (copies/mol) × *wsp* gene concentration (g/μL)/*wsp* gene molecular weight (g/mol) [46].

The data of each population was analyzed by TWOSEX-MS Chart software, and the survival rate (H), fecundity (mx), and other parameters of each population were obtained. Data were expressed as the mean ± standard error, and the data were subjected to one-way analysis of variance (ANOVA)($p < 0.05$) followed by a significant difference test (LSD) using SPSS statistics 19.0 (IBM, USA) and using Sigmaplot 14.0 (Systat International Software, USA) to plot.

## 3. Results

### 3.1. Identification of Symbiotic Bacterium in Tea Green Leafhopper

Separation of the fragments by electrophoresis produced a clear and bright electrophoresis strip. Lanes 1, 2, 3, and 4 represented the samples from Fenggang, Jiu'an, Meitan, and Pingtang, respectively (Figure 1), and a single band was obtained by agarose gel electrophoresis only from PCR products that contained genomic DNA of *Wolbachia*. It was clear that samples collected from Fenggang (lane 1), Jiu'an (lane 2), and Meitan (lane 3) were successfully amplified. The sample collected in Pingtang (lane 4) failed to appear on the PCR, indicating no *Wolbachia* infection in the tea leafhoppers from Pingtang. In addition, the bands of PCR products in lane 1 (Fenggang) and lane 2 (Jiu'an) were both brighter than the band in lane 3 (Meitan), indicating the possibility of high infection rate in leafhoppers from those two locations. However, no bands were detected for the *16S rRNA* and *Ftsz* genes.

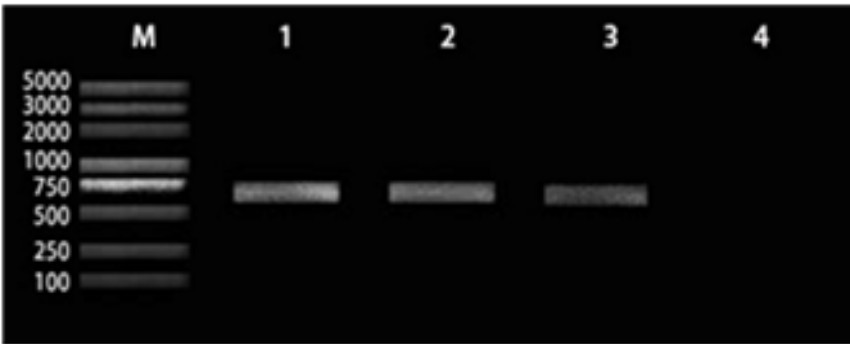

**Figure 1.** The amplification based on the *wsp* gene of *Wolbachia*. 'M' is DNA marker. The numbers of different lanes 1, 2, 3, and 4 denote the samples collected in Fenggang, Jiu'an, Meitan, and Pingtang respectively.

*3.2. Wsp Content in Tea Green Leafhopper by Real-Time PCR*

The standard curve was generated according to the relation between the *wsp* gene concentration in total DNA and cycle threshold (CT) (Figure 2). The regression equation was established as y = −2.864x + 32.51, where $R^2$ = 0.9816 indicates that the concentration of *wsp* gene in the diluted sample had a good linear relationship. Therefore, the content of *Wolbachia* in different samples could be calculated by measuring CT values.

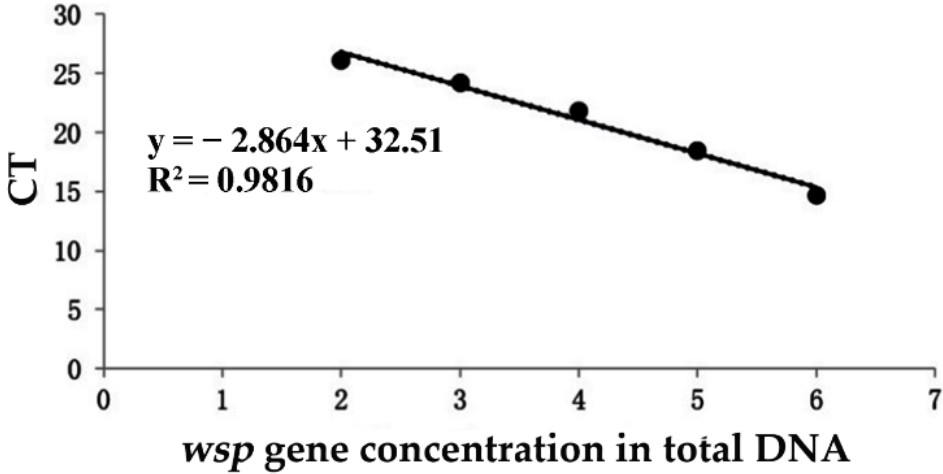

**Figure 2.** The standard curve of *wsp* gene obtained from real-time PCR.

The relative amounts of *wsp* gene to total DNA (*wsp* gene concentration) in various samples, including four different geographic locations, different insect body parts and ages, and from both sexes of tea green leafhopper were determined by real-time PCR. The *wsp* gene concentration in total DNA could be calculated using the standard curve equation from Figure 2 and obtained content of *Wolbachia* in different samples (Figure 3).

As shown in Figure 3A, there was no significant difference for the content of *Wolbachia* in samples from three (Fenggang, Jiu'an, and Meitan) of the four involved locations, with around 3.3–3.7 copies/μL of *wsp* gene copy, while *Wolbachia* could not be detected in the samples from Pingtang. *Wolbachia* levels in different body parts of the female insect (Figure 3B) fit the order of abdomen > thorax > head, with a statistically significant difference observed. In addition, a female tea green leafhopper had a much higher *Wolbachia* amount (4.4 copies/μL) than that of a male (3.1 copies/μL) (Figure 3C). Moreover, *Wolbachia* amount was found to increase from 2nd instar to 5th instar and arrived at the max at 5th instar in the nymph stage and kept increasing in the adult stages (Figure 3D).

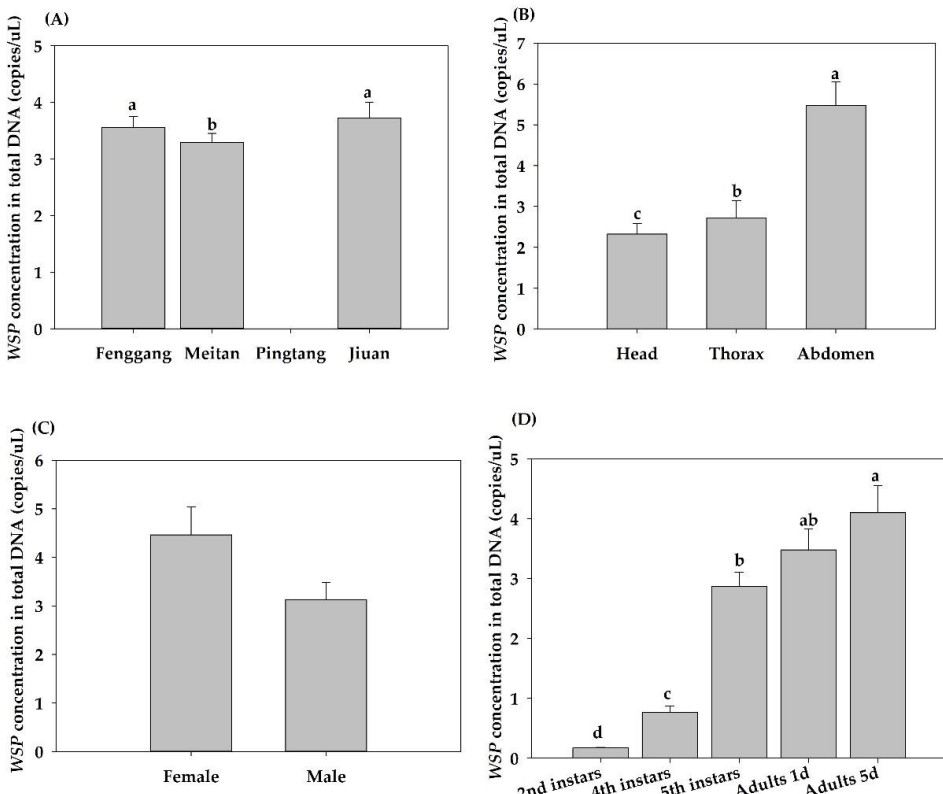

**Figure 3.** The relative amounts of *wsp* gene to total DNA (copies/μL) in different samples of tea green leafhopper. (**A**) Whole adult insects from four geographical locations. (**B**) Body parts of insects in adult stage. (**C**) Whole insect body of female and male. (**D**) Whole insects in different development stages. Different letters indicate significant differences at $p \leq 0.05$.

### 3.3. Localization of Wolbachia by FISH

The nymphs and adults of the tea green leafhopper in the natural population of Jiu'an were in situ hybridized to localize *Wolbachia* (Figure 4A–D). It was obvious that the fluorescence intensity in the figure intensified from diagram A to diagram D, which indicated the *Wolbachia* content in tea green leafhopper increased from 2nd instar to adult stage.

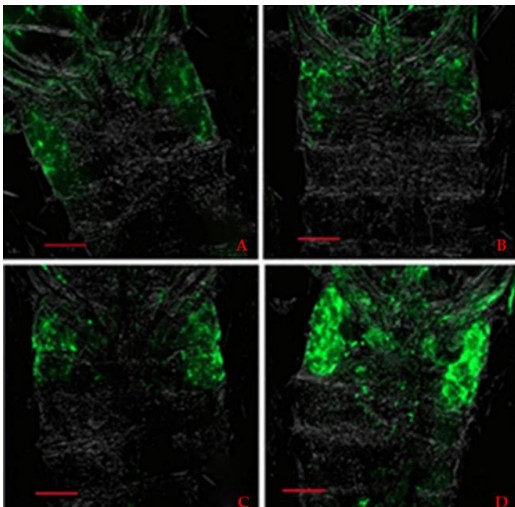

**Figure 4.** FISH on tea green leafhopper at different developmental stages; fluorescence illustration refers: (**A**) 2nd instar), (**B**) 4th instar, (**C**) 5th instar, (**D**) adult 5 days. (Red scale bar = 1 mm).

### 3.4. Elimination of Wolbachia in Tea Green Leafhopper

The cure effect of tetracycline (1, 5, and 10 mg/mL) on *Wolbachia* infection in tea green leafhopper is shown in Figure 5. *Wolbachia* infection continued to decline during observation, and the infection rate in groups 5 and 10 mg/mL tetracycline were completely extinguished by day 25 and day 20, respectively. Meanwhile, the infection rate in the 1 mg/mL treatment group fell to 60% by day 25 and maintained a gradual decline until day 35 or death of the adult. This indicated that tetracycline solution can eliminate *Wolbachia*, and 5 mg/mL is properly effective.

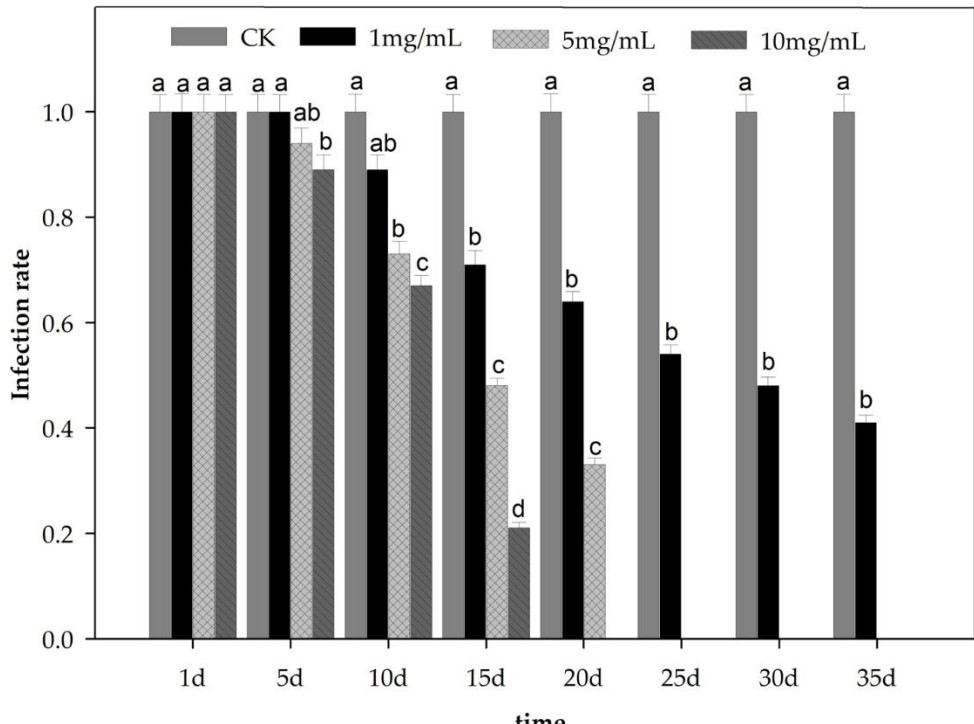

**Figure 5.** Biotics treatment and *Wolbachia* infection rate. Different letters indicate significant differences at $p \leq 0.05$.

### 3.5. Effects of Symbiotic Bacteria on the Reproductive Development of Tea Green Leafhopper

The influence of *Wolbachia* on the insect stage development and total life span of tea green leafhopper was investigated by comparison between natural population and cured population (Figure 6).

As can be seen, stage period was generally prolonged in the cured populations group (Figure 6A,B,D) and hence the total life span was also prolonged (Figure 6E). This led to the total lifespan difference between the natural group and the cured group. Since tea green leafhopper would differentiate into male and female individuals when they emerged to the adult stage, their lifespans differed and apparently females live longer, with 31 days for female and 28 days for male in natural populations and 38 days for female and 34 days for males in cured populations. In sum, the life span of the whole generation of cured populations was prolonged for about 6–7 days longer than the natural population (Figure 6E). Therefore, it could be assumed that the symbiotic bacterium *Wolbachia* had an influence on the growth and development of tea green leafhopper, shortening its generation cycles.

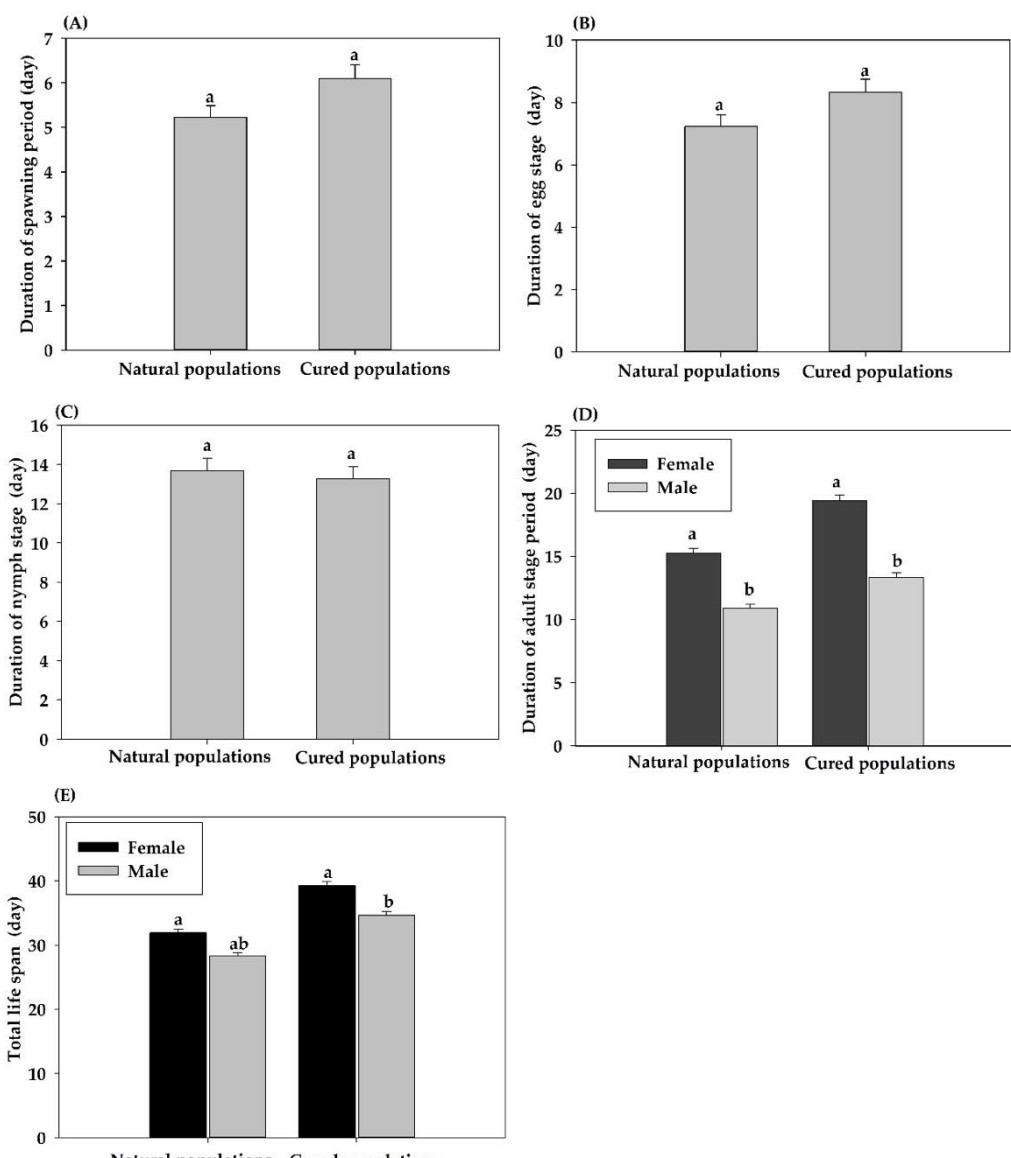

**Figure 6.** Influence of symbiotic bacteria *Wolbachia* on the developmental duration of different stages of the tea green leafhopper: (**A**) spawning period; (**B**) egg stage; (**C**) nymph stage; (**D**) adult stage period; (**E**) total life span. Different letters indicate significant differences at $p \leq 0.05$.

### 3.6. Mating Result between the Wolbachia-Infected Host and Uninfected Tea Green Leafhopper

This combination of partners and their progeny's life information were investigated (Table 1). The leafhoppers' mating was designed as the following four types: W−♀×W♂, W♀×W♂, W♀×W−♂, W−♀×W−♂ (♀female, ♂male). "W" represented naturally infected populations and "W−" represented *Wolbachia*-free (uninfected) populations that were cured by tetracycline. The numbers of the offspring derived from different crossing pairs W−♀×W♂, W♀×W♂, W♀×W−♂, W−♀×W−♂, were 16, 18, 20, and 25, respectively, with significant deference. Obviously, the mates with cured male W−♂ produced relatively more offspring than the mates with naturally infected male W♂.

**Table 1.** Influence of mating types on the offspring number, sex ratio and life span of the tea green leafhopper.

| Mating Type | No. of Observe Mates | No. of Nymph from per Mate | Average Sex-Ratio (Female♀: Male♂) | Average Adult Period of Offspring | |
|---|---|---|---|---|---|
| | | | | Female (♀)/day | Male (♂)/day |
| W−♀×W♂ | 17 | 16 d | 1.7 (10:6) | 13.0 a | 9.8 c |
| W♀×W♂ | 19 | 18 c | 1.6 (11:7) | 9.8 a | 9.5 b |
| W♀×W−♂ | 21 | 20 b | 1.5 (12:8) | 16.2 a | 11.3 b |
| W−♀×W−♂ | 16 | 25 a | 1.3 (14:11) | 14.1 a | 12.7 a |

Note: Letters (a, b, c, d) table indicate the significant difference when $p \leq 0.05$, and the data indicate the mean $\pm$ standard error. 'W' represented the natural infected population of tea green leafhopper. 'W−' represented the cured populations.

We also calculated the ratio between numbers of female and male offspring by sex differentiation at adult stage. The general population deviated from equilibrium by favoring female at 1.7, 1.6, 1.5, and 1.3, respectively. The mate (W−♀×W♂) produced the biggest disequilibrium and was obviously skewed towards dominance of females (1.7:1). For the mates of cured populations (W−♀×W−♂), the number of female and male offspring were close, at 14 female insects and 11 male insects, and sex ratios was less biased (1.27:1, close to 1).

The durations of the adult period for female and male offspring were also observed and recorded (Table 1). As shown, the adult period ranged from 9.5 to 12.7 days for males and 9.8–15.4 days for females. The offspring from mate W♀× W♂ retained the shortest adult lifetime, at 9.5 days for males and 9.8 for females. It is quite clear that the female offspring had relatively longer lifetimes than the males. Also, new generations from mates with *Wolbachia*-infected males (W−♀×W♂ and W♀×W♂) lived relatively short adult periods.

## 4. Discussion

In this study, the symbiotic bacterium *Wolbachia* was examined for its existence and effects on the host tea green leafhopper, based on *wsp* gene (which evolves quickly and is suitable for grouping *Wolbachia* strains [47,48]). According to PCR amplification, *Wolbachia* was present or hosted in the tea leafhopper collected from three tea-production locations (Jiu'an, Meitan and Fenggang) but absent from samples from Pingtang. The result might be caused by different altitude, climate, or ultraviolet intensity in different locations [49,50]. On the one hand, possible extremely low *Wolbachia*-infection rate among populations made it impossible to detect genes using PCR molecular detection. Another reason might be the random samples collection falling outside the area of highly infected clusters [51]. Also, assured by real-time PCR, *wsp* gene content differed among those hosts and increased with the evolution of tea leafhoppers. The FISH study verified the result and was consistent with the real-time PCR detection in terms of insect stage development comparison. Later, we used a double-negative control (no-probe and RNAse-digested control) as suggested by Nugnes et al. [45] and did not find any signal (data was not provided).

*Wolbachia* could be removed or cleared from insects by tetracycline and rifampicin antibiotics [52,53]. Some studies explored the effect of antibiotics on the removal of *Wolbachia* in different insects [54,55]. This process might not only involve bacterial clearance but also other changes in the microenvironment of the host [56]. Many studies reported that *Wolbachia* had obvious effects on the host's reproductive development through multiple mechanisms, mainly including CI, male killing, parthenogenesis, feminization, and enhancement of female reproduction and male fertility, etc. [25–27,57–59].

CI meant some hosts infected with *Wolbachia* strains showed early embryogenesis death when *Wolbachia*-infected males mated with uninfected females [28,29]. In this study, mating of infected male and uninfected females resulted in a tendency of CI, with the highest being 25 offspring from the one pair uninfected mate W−♀×W−♂ and the lowest being 16, derived from the crossing pair of W−♀×W♂. Also, *Wolbachia* infection caused a female-biased sex ratio.

The inherited symbiotic bacteria play an important role in manipulating the reproductive biology of their hosts. Therefore, the use of insect symbiotic bacteria for developing new pest control technologies has important application prospects [60].

## 5. Conclusions

In this study, the presence and influence of symbiotic bacterium *Wolbachia* was investigated in tea green leafhoppers, the main pest in tea gardens. *Wolbachia* presence could manipulate evolution of tea green leafhopper hosts by inducing changes in host reproductive and sex-determination systems. The frequent incidence of *Wolbachia* and their influence has led to the proposal of their use as potential agents in the evolution and control of tea pests.

**Author Contributions:** Q.Z. conducted the experiments; Q.Z., R.L., D.J., Y.T. and X.T. designed and performed the experiments and sample collection; Q.Z., R.L., Y.T., X.Z. and Q.W. analyzed the data; L.J. conceived and supervised the project and wrote the manuscript. All authors have read and agreed to the published version of the manuscript.

**Funding:** This research was funded by science and technology project of Guizhou province, [2015]5020 and scientific research projects of major agricultural industries of Guizhou province, [2019]006.

**Institutional Review Board Statement:** Not applicable.

**Informed Consent Statement:** Not applicable.

**Data Availability Statement:** No other data supporting report available.

**Conflicts of Interest:** The authors declare no conflict of interest.

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
