# Peer review of "The Detection of Wolbachia in Tea Green Leafhopper (Empoasca onukii Matsuda) and Its Influence on the Host"

_agriculture, doi:10.3390/agriculture12010036_

Round 1

Reviewer 1 Report

The present version of the manuscript is much improved compered with the initial submission. Most of the changes we suggested have been addressed. Nevertheless, some points within the text need to be clarified clearly (they are stated in details in the attached pdf file). Moreover, further linguistic and syntax revision is needed.

Author Response

Cover letter

Dear reviewer, we are so appreciated your scrupulous review and great advice. We have taken care of all the revisions accordingly.

Reviewers 1

L12 Delete “pests”

Responses: Done.

L15 When you write the name 'tea green leafhopper' dont use plural in leafhopper. Change it in the rest of the article also

Responses: Thanks for your valuable comments and suggestions. We have revised it in the manuscript.

L31 “,”

Responses: Done.

L32 Delete “is an”, “insects”

Responses: Done.

L39 “in have”

Responses: Done. “in have” was changed to “has”.

L40 “leafhopper”

Responses: Done. “leafhopper” was changed to “leafhoppers”

L42-43 Inhibit the bacteria??? i think the opossite. you want the bacteria to control the insect

Responses: Yes, we have modified it to " It is expected to be an ecofriendly alternative management method to control tea leafhopper by inhibiting or antagonizing symbiotic bacteria and realize the green and sustainable development of tea production." Thanks!

L47 forgot to open the bracket " [ "

Responses: Done.

L48 “could”

Responses: Done. “could” was changed to “can”.

L59 “giving birth to”, “so the number of insects is reduced”

Responses: Done. “result in” was changed to “giving birth to”, “and reduce the number of insects.” was changed to “and the number of offspring is reduced.”.

L61 “, its different”

Responses: Done.

L64 L66 neither here needs parenthesis

Responses: Done.

L69 Delete “would”

Responses: Done.

L71 “potential”, delete “measure on insect vectors”

Responses: Done.

L73 Delete “a”

Responses: Done.

L75 one word

Responses: Done.

L80 doesn't need capita

Responses: Done.

L87 “province” “,”

Responses: Done. “Provience” was changed to “province”.

L87 you mean average altitude?

Responses: Yes, the four places where the samples were collected had little difference in altitude, so we used average altitude to record them.

L90 “. Some”

Responses: Done. “, some” was changed to “. Some”

L92 leafhopper's eggs which was adhered to, Delete “collected in Jiu'an”

Responses: Done. “leafhopper's eggs adhered to fresh” was changed to “leafhopper's eggs which was adhered to”.

L93 “collected in Jiu'an and”

Responses: Done. “Collected” was changed to “collected in Jiu'an and”.

L95 “which was changed”, “and used to ensure”

Responses: Done. “which was changed ” was changed to “which was replaced”, “used to ensure the freshness of the tea branches” was changed to “ for keeping the tea branches fresh”.

L105 “The whole” “grounded”

Responses: Done. “Whole” was changed to “The whole”, “ground” was changed to “grounded”.

L108 Delete “in the”, “used in”

Responses: Done.

L109 “buffer was used.”, “The concentration was determined by a Nanodrop”

Responses: Done. “buffer” was changed to “TE buffer was applied.”, “concentration determined using a Nano Photometer(IMPLEN)” was changed to “The concentration of elution was determined by a Nanodrop(IMPLEN)”.

L117 “dNTP's”

Responses: Done. “dNTP” was changed to “dNTP's”

L175 “the number was recorded”

Responses: Done. “Recorded the number” was changed to “the number was recorded”.

L176 “moisturized”

Responses: Done. “moisturize” was changed to “moisturized”

L177 “???”, “observed”

Responses: Done. “read” was changed to “reared”, “observe” was changed to “observed”.

L178 “something missing. the whole sentence doesnt make sense”

Responses: Thanks for your valuable comments and suggestions. We have modified to “the newly emerged nymphs were reared with fresh tea buds and observed every 12 hours for its reproductive parameters such as the time span of each stage(spawning, egg, nymph, and adult stages) until their death.”

L181 you need to close the extgra parenthesis too

Responses: Done.

L191 “were”

Responses: Done. “are” was changed to “were”.

L199 “respectively”

Responses: Done.

L205 the line 1 is as bright as the line 2. isnt there any possibility of high infection? why dont you mention it?

Responses: Thanks for your valuable comments and suggestions. We have explained in the manuscript. Separation of the fragments by electrophoresis indicated clear and bright electro-phoresis strip. Lane 1, 2,3 and 4, represented the sample from Fenggang, Jiu’an, Meitan and Pingtang respectively (Figure. 1) and a single band was obtained on agarose gel electrophoresis only from PCR products that contained genomic DNA of Wolbachia. It was clear that samples collected from Fenggang (lane 1), Jiu’an (lane 2) and Meitan (lane 3) were successfully amplified. The sample collected in Pingtang (lane 4) failed to appear on PCR indicating no Wolbachia infection in those leafhopper from Pingtang. In addition, the band of PCR products in lane 1 (Feng gang) and lane 2 (Jiu’an) was both brighter than the band in lane 3(Meitan) indicating the possibility of high infection rate in leafhopper from those two location. However, no bands were detected for the 16S rRNA and Ftsz genes.

L233 This is not in agreement with the figure 4, which shows that thorax of the adult (fig 4.4) has more Wolbachia than the abdomen (fig. 4.6). Please clarify.

Responses: Thanks for your valuable comments and suggestions. The Wolbachia level in abdomen was higher than that in chest of same individual insect. But here we included samples from different periods and different organ make the comparison confused. To avoid errors, we removed picture 5 and 6 in Fig. 4.

L246 “increases”

Responses: Done. “increase” was changed to “increases”

L258 without fullstop (.) and the higher concentration? or you want to conclude that even 5 is still good enough

Responses: Yes, we have revised according to your suggestion.

L279 “duration of”

Responses: Done. “duration” was changed to “duration of”

L320 “has (here you need simple present)”

Responses: Done.

L324 “for developing”

Responses: Done. “to develop was changed to “for developing”

L326 “was (when you say bacterium is singular)”

Responses: Done.

L337 “especially”

Responses: Done. “specially” was changed to “especially”

L340 “Also”

Responses: Done. “And” was changed to “Also”.

L341 “were increased as the developmental stages increased”

Responses: Done. “increased in tea green leafhopper with an increase in insect instars” was changed to “were increased as the developmental stages increased”.

L345 ??? was it?

Responses: Thanks for your valuable comments and suggestions. We responded in L233. Thanks.

L361-362 This point needs a clear and extended explanation. Moreover, this claim contrasts line 233, in which you refer that Wolbachia level ....  fit the order of abdomen > thorax > head... Please clarify

Responses: Thanks for your views and comments. We have clearly described in the manuscript.

L381 Delete “and.”

Responses: Done.

Reviewer 2 Report

The manuscript reports the infection rate, distribution, and the effect on reproduction of host for a strain of Wolbachia infected in Empoasca onukii. The data shown in the manuscript is valuable as the natural history of this pest species. However, a part of the research methods such as statistical analyses or the infection status of insects used for the experiment is not shown sufficiently. In addition, several sentences can cause a misunderstanding. L21 This sentence may be problematic. There is no sufficient evidence for feminization in this species. L81 The word “in” should be italic. Fig.3 Although the authors used the term “biomass” in Fig. 3, the biomass was not measured directly. I think the term is not proper. The authors should take care that the value was estimated indirectly. L214 What are the plasmids? Materials & Method did not explain it. What is the source of these plasmids? Fig.5 & Fig.6 What is CK in these figures? Please explain in the caption. In addition, I do not understand the reason why the maximum value of infection rate is 1 but not 100, although the symbol % was used. L275 The authors pointed out that Wolbachia has no effect on the egg and nymphal stages but not the adult stage. However, I cannot understand the statistical analysis used for the data in Fig.7. Did the authors pool any data shown in Fig. 7? It is obvious that the number of days for development among the egg, nymph, and adult stages. I think that the comparison between two groups within a stage is suitable. L275 I can not understand the status of insects from tetracycline-treated populations. Were they the offspring of females treated by tetracycline? Fig.8 I cannot understand the statistic analysis again. In the group of W♀×W-♂ and W-♀×W♂, the bars had the letter of “b”. However, the values of these bars were intermediate to other bars with the letter of “a”. I recommend the authors use the regression analysis or GLM despite the complicated analysis used in this manuscript. These analyses can show the effect of the infection status of males and females clearly. L312 Why CI observed in this study was incomplete? Could Wolbachia remain in insects of tetracycline-treated populations? Did the authors check the infection status of insects used for mating experiments by the PCR method? L318 I cannot agree with the sex ratio of W-♀×W♂ group caused by feminization or male-killing. If so, why did W-♀×W-♂ group show a low sex ratio? L330 This sentence may be problematic. There is no sufficient evidence for feminization in this species. L388 This sentence may be problematic. There is no sufficient evidence for feminization in this species.

Author Response

Response to reviewer 2

Dear reviewer, we are so appreciated your scrupulous review and great advice. We have taken care of all the revisions accordingly.

L21 L330 L388 This sentence may be problematic. There is no sufficient evidence for feminization in this species.

Responses: Yes we agree with you. Thanks for your valuable comments and suggestions. we have changed "feminization or male-killing" to "Sex ratio imbalance".

L81 The word “in” should be italic.

Responses: Done.

Fig.3 Although the authors used the term “biomass” in Fig. 3, the biomass was not measured directly. I think the term is not proper. The authors should take care that the value was estimated indirectly.

Responses: Thanks for your valuable comments and suggestions. We have replaced "biomass" with "content", which represents the concentration or amount of WSP gene in the total DNA.

L214 What are the plasmids? Materials & Method did not explain it. What is the source of these plasmids? Fig.5 & Fig.6 What is CK in these figures? Please explain in the caption. In addition, I do not understand the reason why the maximum value of infection rate is 1 but not 100, although the symbol % was used.

Responses: Thanks for your valuable comments and suggestions. In this experiment, the Plasmids represent the concentration of the wsp gene in the total DNA, which we have modified in Fig. 3. In Fig.5 and 6, CK represents clean water without tetracycline; at first, 1 represents 100%. In addition, we have modified the digital expression of the chart according to your suggestions.

L275 The authors pointed out that Wolbachia has no effect on the egg and nymphal stages but not the adult stage. However, I cannot understand the statistical analysis used for the data in Fig.7. Did the authors pool any data shown in Fig. 7? It is obvious that the number of days for development among the egg, nymph, and adult stages. I think that the comparison between two groups within a stage is suitable.

Responses: Thanks for your valuable comments and suggestions. As you pointed out that the analysis of Fig. 7 is not clear, we have redrawn the diagram as follows:

L275 I can not understand the status of insects from tetracycline-treated populations. Were they the offspring of females treated by tetracycline? Fig.8 I cannot understand the statistic analysis again. In the group of W♀×W-♂ and W-♀×W♂, the bars had the letter of “b”. However, the values of these bars were intermediate to other bars with the letter of “a”. I recommend the authors use the regression analysis or GLM despite the complicated analysis used in this manuscript. These analyses can show the effect of the infection status of males and females clearly.

Responses: Thanks for your valuable comments and suggestions. This section was recognized and combined data in original Fig 8 to Table 1. And fig 8 were deleted.

Table 1 Influence of mating types on the offspring number, sex ratio and life span of the tea green leafhopper

Mating type

No. of observe mates

No. of nymph from per mate

Average sex-ratio

(Female♀: Male♂)

Average adult period of offspring

Female (♀)/day

Male (♂)/day

W-♀×W♂

17

16 d

1.7 (10:6)

13.0 a

9.8 c

W♀×W♂

19

18 c

1.6 (11:7)

9.8 a

9.5 b

W♀×W-♂

21

20 b

1.5 (12:8)

16.2 a

11.3 b

W-♀×W-♂

16

25 a

1.3 (14:11)

14.1 a

12.7 a

Note: Letters (a, b, c, d) table indicate the significant difference when P≤0.05, and the data indicate the mean ± standard error. ‘W’ represented the natural infected population of tea green leafhopper. ‘W-’ represented the cured populations.

L312 Why CI observed in this study was incomplete? Could Wolbachia remain in insects of tetracycline-treated populations? Did the authors check the infection status of insects used for mating experiments by the PCR method?

Responses: Most insects have Wolbachia in their bodies, but due to their small amount in the host, the effect on the host is not obvious. In this study, the samples we used were natural populations and populations treated with tetracycline, so our results only showed a trend of CI, and did not cause complete CI. In addition, during the experiment, we have experimental PCR to detect the effect of tetracycline treatment.

L318 I cannot agree with the sex ratio of W-♀×W♂ group caused by feminization or male-killing. If so, why did W-♀×W-♂ group show a low sex ratio?

Responses: Thanks for your valuable comments and suggestions. Sure your are corrected. We made the wrong description like “sex ratio of W-♀×W♂ group caused by feminization or male-killing”. According to the research results, the presence of symbiotic bacteria caused the change and imbalance of sex ratio. There is no clear evidence for male killing and feminization(In fact, male killing should be the reason of female-basis ratio). We have changed "feminization or male-killing" to "sex ratio imbalance".

This manuscript is a resubmission of an earlier submission. The following is a list of the peer review reports and author responses from that submission.

Round 1

Reviewer 1 Report

This manuscript tries to describe the detection of Wolbachia in tea green leafhopper (Cicadellidae) and its influence on the host. Although it is an interesting idea, there are too many linguistic errors, which make reading of the manuscript difficult... I tried to correct them, but extensive linguistic review is required. Moreover, I have spotted some other corrections, which are included in the attached pdf file.

Author Response

Thanks for your kind dedicated work. All your suggestions have been carefully taken care of. And the English writing was polished and increased.

Point 1: L14 the “tea green”

Responses: Done. We added “tea” in front of “green leafhoppers”

Point 2: L29 Empoasca onukii Matsuda”

Responses: Done. We added “Matsuda” to “Empoasca onukii Matsuda”.

 Point 3: L30 “because of its”

Responses: Done. “With the characters of” was changed to “because of its”.

 Point 4: L31 “and high mobility”, “is”, “it”.

Responses: All has been taken care of accordingly.

 Point 5: L32 “It has an hemimetabolous development insects”. “serious generations overlap with”.

Responses:  Done. “And it is an hemimetabolous insects” was changed to “It has an hemimetabolous development insects”. And “serious generations overlap with” was deleted.

Point 6 L34 “causes”,  “to wither and become brown”. Delete “of tea green leafhopper”

Responses: Done.  “cause” was changed to “causes”.  “Withered and brown” was changed to “to wither and become brown”.

 Point 7: L34 “controlling method”, “use”.

Responses: Done. “Current controlling” was changed to “controlling method” and “utilization” was changed to “use”.

 Point 8: L38 “of”

Responses: Done.

Point 9: L39 “that”  “is capable to”

Responses: It has been taken care of according to your suggestions.

 Point 10: L45 “interacting”

Responses: Done.

 Point 11: L46 “Some examples of endosymbiosis”

Responses: “Examples” was changed to “Some examples of endosymbiosis”

 Point 12: L48 “of intracellular bacteria”

Responses: “of bacteria” was changed to “of intracellular bacteria

Point 13: L49 “A study or studies”

Responses: Thanks.  We changed it as “Studies”.

Point 14: L55  “have”, “effect”

Responses: Yes and it is done according to your suggestions. “has” changed to “have”、“action” was changed to “effect”

 Point 15: L56 “Also”

Responses: Done

Point 16: L57 It is like you say it reproduce with CI of females and not that it reproduce with the female that isnt infected with CI. please rephrase it

Responses: Thanks. The last sentence has been revised as “CI occurs when a Wolbachia infected male mates with an uninfected female and can cause embryonic lethality and reduce the number of insect vectors. An infected female is compatible with any infected or uninfected male of the same Wolbachia strain [26,27].” And a reference 27 was added.

 Point 17: L58 “reduce”

Responses: Done.

 Point 18: L58 “insect vectors”

Responses: Done  

 Point 19: L59 “nowadays”

Responses: Done.

Point 20: L61 the “trm eat”

Responses: “trm eat” was revised as “treat”.

 Point 21: L62 “the influence” “hosts'”

Responses: Done.

Point 22: L66 its better to say: that in En. formosa offspring which have been treated with tetrac. hydroc....

Responses: Thanks. We revised it as “that when whitefly parasite (Encarsia formosa) was treated by tetracycline hydrochloride, its offspring including nymph and black pupae would lessen.”

Point 23: L67 “was significantly increased”

Responses: This paragraph was revised as above response.

 Point 24: L67 you mean researchers?

Responses: We actually want to describe the research’s work. it was should not be vague like this. So we combined the sentence and modified it as following.

“The report concluded that the symbiotic bacteria Wolbachia has a negative impact on the reproductive development of Encarsia formosa.”

Point 25: L69(1)you mean insect vectors that spread viruses?? what viruses?? you mean generally? if yes write it correctly;(2)confirms (if the subject is "Research")

Responses: Thanks for your valuable comments and suggestions.  We rewrote this sentence as” The use of Wolbachia as a potential biological control measure on vector insect has been reported [36,37]. For example,Wolbachia decreased mosquito spermathecae and inhibited the leather fever virus (DENV) and Zika virus (ZIKV) infection [36]. Wolbachia infection in small brown planthopper (Laodelphax striatellus) could reduce its transmitting of rice stripe mosaic virus (RSV) [37].”

Point 26: L69 “to control”

Responses: Done.

Point 27: L71 “and potentially can be used”、 “that they”

Responses: Yes and it is done according to your suggestions.

“Those reports confirmed that Wolbachia was a benign bacterium and potentially can be a used as friendly biological agent to control number of pests and diseases”.

Point 28: L73 “investigate”

Responses: Done.

Point 29: L80 only adults?? not nymphs??

Responses:  Thanks. Yes, we only collected adults in one hand it was directly used in DNA extraction. And on the other hand, adults were easy reared to obtain new insects’ generation and testing targets

 Point 30: L82 subjected to tests of the genes used directly in the sample???

Responses: The meaning expressed here is that the sample was directly used for DNA extraction and gene detection of the presence of Wolbachia in the insect. So we modified it as “directly used for gene detection of Wolbachia's distribution”.

 Point 31: L79 “collection”

Responses: Thanks, and it is done accordingly.

 Point 32: L80 “surrounding areas”

Responses: Done

 Point 33: L84 “continued” 、“under”.

Responses: Done.

Point 34: L85 “capital”

Responses: Yes. It is change as “Fresh”

Point 35: L88 you mean the insect rearing?

Responses:  Yes, it means the insect rearing method. We modified the sentence in line 86-88 as “This insect rearing method was involved in the study of Wolbachia’s effects on the reproductive development of tea green leafhoppers”.

 Point 36: L86 which was used for”

Responses: Done.

Point 37: L87 “to”、 “study about”

Responses: Done. This sentence was rewrote.

 Point 38: L93 without any other solution?

Responses:  Thanks for your valuable comments and suggestions.

Sure there is solution of liquid nitrogen and ddH2O.

 Point 39: L96 it would be good to be a little more specific about the procedure of DNA extraction.

Responses:  We modified this expression as “Sample preparation:For each sample, five adults of tea green leafhopper were placed into the bottom of a 1.5 mL microcentrifuge tube containing liquid nitrogen and grind it into powder with a tissue grinder. Then added 200 μL ddH2O and 20 μL Protenase K into the tube and vortex the tube for 10 s. Then add 200 μL Buffer gA and incubate the tube at 56°C for 1 h. After that, 200 μl of absolute ethanol were added to the tube and the solution were vortexed and mixed well.

DNA binding: the mixture obtained in above steps was transfer to a spin column. centrifuged at 12000×g for 1min and the liquid was discarded.

Washing: Then add 500 μL Buffer PW to the Spin column and centrifuge the spin column at 12,000×g for 30s, discard the waste liquid. And repeat this step once again with adding 500 μL Wash Buffer.

Elution: Then put the spin column back into the collection tube and centrifuge at 12000×g for 2min, and then dry for 1 min by open the lid. Take out the Spin Column again and put it in a new 1.5mL centrifuge tube, add 55 μL TE Buffer to the center of the adsorption membrane, and store it at 20-25℃ for 2 min. Centrifuge at 12000×g for 2 min to obtain DNA which was the stored at -20 ℃.”

Point 40: L97 “was then stored”

Responses: Done.

 Point 41: L100 didn’t you use MgCl2 ???

Responses:  Thanks. The kit for PCR extraction already contains all the reagents that need to be tested, so no extra agents needed.

Point 42: L115 you mean adults and nymphs? because you said that you collected adults.

Responses: Thanks. We collected adults and eggs attached to tea branches from the field. We use the method described in 2.1 to hatch and rear the insects.

Point 43: L115 “from both sexes”、 “for detection of symbiotic bacteria”

Responses: Yes and it is done according to your suggestions. “male and female individuals” was changed to “from both sexes” and the “symbiotic bacteria detection” was changed to “for detection of symbiotic bacteria.

Point 44: L117 did you use the same protocol for DNA extraction here?

Responses: Thanks. Yes, we used the same protocol. The only difference is that more individual insects involved in each sample of PCR.

 Point 45: L120 600 S?

Responses: Thanks for your valuable comments and suggestions. 600 S means the time set by the test instrument Roche LightCycler 96, which is determined by the thermal sensitivity of the device itself.

 Point 46: L147 in results you say you use 10 mg/ml also.

Responses: We are sorry for the missing information.  We used three concentrations in including 1 mg/mL, 5 mg/mL and 10 mg/mL. and it has been taken care of.

 Point 47: L148 “using”, “solution were smeared on”

Responses: Done.

 Point 48: L149 “we dried” “subsequently”

Responses: Done.

 Point 49: L150 “nymphs were taken and put”?????

Responses: It was revised as “nymphs were transferred into the fresh tea leaves.”

Point 50: L151 “Tetracycline was applied to the”、 “for” “was applied to”

Responses: Done.

 Point 51: L153-156 Please rephrase it. it isn’t very clear. Try to use better the verbs in the sentence

Responses: Thanks for your valuable comments and suggestions. We revised it as following “The tea green leafhoppers were then treated with different concentrations of tetracycline and dynamically observed 35 days every five days.”

 Point 52: L158-170 there are many mistakes in this paragraph. you write: let it lay, replace, record. its like reading a protocol. not describing the protocol you use. it’s better to say eg. the tea buds were replaced and not replace the teabuds. all the verbs have to change into the correct for.

Responses: Thanks. We rewrote this paragraph.

The naturally Wolbachia infected populations and uninfected (tetracycline-treated populations) were compared for their longevity in different stage.

 Newly emerged adults (two pairs) are randomly selected and raised in self-made vials, at 25 ±1 ℃, L/D=16:8, and RH 65±5 %. 15 replicates were set for each treatment. The fresh tea buds were provided and replaced every day. The eggs of tea green leafhopper attached to tea buds were collected and recorded the number. Those eggs were then gathered in an artificial climate box and moisturize with wet filter paper until they hatch. After hatching, the newly emerged nymphs were read with fresh tea buds observe every 12 hours. and reproductive parameters such as the longevity of early spawning, egg, nymph, and adult stages until their death.

 Point 53: L184 “and difference”

Responses: Done.

Point 54: L193 “respectively”, “pingtang respectively”

Responses: Yes and it is done according to your suggestions.

Point 55: L195 “while”.

Responses: Yes and it is done according to your suggestions.

Point 56: L199 it seems that both line 1 and 2 are brighter from 3. if its only line two correct "the bands" to "the band" and "were brighter" to "was brighter"

Responses: Done

Point 57: L204 "italics"

Responses: Done

Point 58: L216 "different"

Responses: Done

Point 59: L229 discussion??

Responses: Thanks. This description seemed like discussion and should be arranged in the discussion part as revised.

Point 60: L237 include in your description which are the different developmental stages and which is female and male for the abdomen.

Responses: Thanks for your valuable comments and suggestions. It has been revised as in the graph.

Point 61: L239-247 Are the results consistent to the comparison between chest and abdomen? because in fish results (in the photos) it seems that there is more bacteria in chest

Responses: Thanks for your valuable comments and suggestions.

With careful analysis, we agree with you in terms of photo 4. So we discussed this result in line 382-384 “It was obvious that the FISH result was consistent to the PCR detection in terms of insect stage development and between gender comparison. However, it seemed that there was brighter fluorescence expression in chest (5-days- adult, fig 4-4) than in the abdomen part (1-day-old adult)”.

Point 62: L256 “italics”, “Also”, higher”

Responses: Done.

Point 63: L261 “that at”

Responses: Done.

Point 64: L261 “rate”

Responses: “rated” was changed to “rate”.

Point 65: L261 “have”

Responses: Done.

Reviewer 2 Report

Dear authors, please find attached my pdf with comments.

The paper needs some improvements especially in the research design because some crucial aspects (other infection, auto-fluorescence, phylogenetic analysis) were treated without the right precision. The concept is interesting, but paper shows a plenty of mistakes.Hence, based on my comments, I suggest a major revision.

Kind regards

Here are some papers to take into account:

Wolbachia treatment:

Li, Y. Y., Floate, K. D., Fields, P. G., & Pang, B. P. (2014). Review of treatment methods to remove Wolbachia bacteria from arthropods. Symbiosis, 62(1), 1–15. https://doi.org/10.1007/s13199-014-0267-1

Nakamura, Y., Yukuhiro, F., Matsumura, M., & Noda, H. (2012). Cytoplasmic incompatibility involving Cardinium and Wolbachia in the white-backed planthopper Sogatella furcifera (Hemiptera: Delphacidae). Applied Entomology and Zoology, 47(3), 273–283. https://doi.org/10.1007/s13355-012-0120-z

Zhou, Y., Liu, X., & Yang, Z. (2019). Characterization of terpene synthase from tea green leafhopper being involved in formation of geraniol in tea (Camellia sinensis) leaves and potential effect of geraniol on insect-derived endobacteria. Biomolecules, 9(12). https://doi.org/10.3390/biom9120808

Multilocus and FISH techniques

Baldo, L., Hotopp, J. C. D., Jolley, K. A., Bordenstein, S. R., Biber, S. A., Choudhury, R. R., Hayashi, C., Maiden, M. C. J., Tettelin, H., & Werren, J. H. (2006). Multilocus sequence typing system for the endosymbiont Wolbachia pipientis. Applied and Environmental Microbiology, 72(11), 7098–7110. https://doi.org/10.1128/AEM.00731-06

Gualtieri, L., Nugnes, F., Nappo, A. G., Gebiola, M., & Bernardo, U. (2017). Life inside a gall: closeness does not favour horizontal transmission of Rickettsia between a gall wasp and its parasitoid. FEMS Microbiology Ecology, 93(7). https://doi.org/10.1093/femsec/fix087

Nugnes, F., Gebiola, M., Monti, M. M., Gualtieri, L., Giorgini, M., Wang, J., & Bernardo, U. (2015). Genetic diversity of the invasive gall wasp Leptocybe invasa (Hymenoptera: Eulophidae) and of its Rickettsia endosymbiont, and associated sex-ratio differences. PLoS ONE, 10(5). https://doi.org/10.1371/journal.pone.0124660

Author Response

Responses to the Reviewer 2:

Dear reviewer, thanks for your kind dedicated work. All your suggestions have been carefully taken care of. And the English writing was polished and increased.

 Point 1: L54 “male-killing”.

Responses: Done.  “male death” was changed to “male-killing”.

 Point 2: L55 “sex-ratio”.

Responses: Done

 Point 3: L56-57 The concept of this sentence is partially right, I would suggest explaining better. Please read and refer to “Cytological analysis of cytoplasmic incompatibility induced by Cardinium suggests convergent evolution with its distant cousin Wolbachia” by Gebiola et al., 2017 Proceedings of the Royal Society B. doi: 10.1098/rspb.2017.1433.

Responses: Thanks for your valuable comments and suggestions. According to the references you provided, we have modified this sentence “CI occurs when a Wolbachia infected male mates with an uninfected female and can cause embryonic lethality and reduce the number of insect vectors. An infected female is compatible with any infected or uninfected male of the same Wolbachia strain [26,27]”. And we sure add the recommend reference in 27.

Point 4: L61 not clear, maybe treat?

Responses: Thanks and it is done.

 Point 5: L64 Please cite some papers where wolbachia was found in E. onukii. Is this the first attempt? If yes, authors should have excluded other symbiotic bacteria.

Responses: Thanks for your valuable comments and suggestions. In other leafhoppers have been reported in the presence of wolbachia, but the specific mechanism of action is rarely reported. This is the first report on wolbachia detection in tea green leafhoppers. The following is our list of literatures on leafhoppers.

(1) Mitsuhashi W, Saiki T, Wei W, Kawakita H, Sato M. Two novel strains of Wolbachia coexisting in both species of mulberry leafhoppers [J]. Insect Molecular Biology. 2002;11(6). :577-584. https://doi.org/10.1046/j.1365-2583.2002.00368.x.

(2) Wangkeeree J, Tewaruxsa P, Roddee J, Hanboonsong Y, Martin O. Wolbachia (Rickettsiales: Alphaproteobacteria) Infection in the Leafhopper Vector of Sugarcane White Leaf Disease [J]. Journal of Insect Science. 2020;20(3).  DOI: 10.1093/jisesa/ieaa053.

Point 6: L80-89 I would suggest to use sometimes common name and scientific name of the studied leafhopper.

Responses: Thanks for your valuable comments and suggestions. We have modified according to your suggestion.

Point 7: L88 “not clear, please change”

Responses: Thanks for your valuable comments and suggestions. We revised it as “This insect rearing method were involved in the study of Wolbachia’s effects on the reproductive development of tea green leafhoppers”.

 Point 8: L93 5 adults= 1 sample? How the authors understand if the adults are all infected?

Responses: Thanks for your valuable comments and suggestions. Yes, we select the collected samples by selecting different geographical locations. During the experiment, one head of adults could not be detected with low infection indicator. So we enlarged the number of individual into 5 heads. We checked the general Wolbachia infection without the exact infection rate. 

Point 9: L99 it is not clear why the authors studied only wsp gene. In “Multilocus sequence typing system for the endosymbiont Wolbachia pipientis” by Baldo et al, 2006, in Appl. Environ. Microbiol.  Authors suggested the use of different genes to identify Wolbachia strains.  In my opinion this is a crucial methodology to be applied in this kind of studies.

Responses: Thanks for your valuable comments and suggestions. In the early stage, we tested 16S, FtsZ, and wsp genes as shown in the following graph (date was not included in the research). Neither 16S nor FtsZ detected bands. Only the wsp gene detected the corresponding bands. As shown below:

Point 10: L125 I would suggest to use italics for latin terminologies such as “in situ”. Please check the whole manuscript.

Responses: Thanks. We have changed all "situ" in the full text to italics.

 Point 11: L124-139 Was this probe developed in this work? If so, I think authors should describe the used methodology. Furthermore I would suggest to use a double negative control (no-probe and RNAse-digested control) to exclude the autofluorescence of the leafhopper tissues as in the following papers  

- Life inside a gall: closeness does not favour horizontal transmission of Rickettsia between a gall wasp and its parasitoid. By Gualtieri et al., 2017 in FEMS

- Genetic diversity of the in- vasive gall wasp Leptocybe invasa (Hymenoptera: Eulophidae) and of its Rickettsia endosymbiont, and associated sex-ratio differences. By Nugnes et al., 2015 in PLoS One.

Responses: Thanks for your valuable comments and suggestions.

The probe W2(5′-CTTCTGTGAGTACGTCATTATC-3′) in the study referred the report [39] where has been used as the probe for the detection of Wolbachia in a leafhopper (Yamatotettix flavovittatus Matsumura) of sugarcane.

And sorry for the ambiguous description. We did use a double negative control (no-probe and RNAse-digested control) to exclude the autofluorescence of the leafhopper tissues. And the negative control test which did not display any signals (data were not given), confirming the specificity of the detected signals.

 We revised this part as following “For the FISH study, we used a double negative control (no-probe and RNAse-digested control) as reported in Gualtieri et al.'s research [52]. Likewise, the result of the test did not show any signal (no data was given).

 Point 12: L136 Line 183, 1:1?

Responses: Yes. It was set at 1:1 in volume for the absolute ethanol and PBS solution, we revised it as “absolute ethanol and PBS (1:1, v/v).

 Point 13: L144 wsp---Wolbachia?

Responses: Thanks for your valuable comments and suggestions. “wsp” was changed to “wolbachia” in that line.

Point 14: L145-152 I think authors should have check if tetracycline was absorbed by plants. Leafhoppers are sapsuckers, so is this a new method, simplier to use. With aleirodidae usually a parafilm membrane was used to fed with flies with tetracycline solution, or the roots of host plant were immersed in a solution containing antibiotics. Please refer to the following paper and its bibliography “Review of treatment methods to remove Wolbachia bacteria from arthropods “by Li et al., 2014 in Symbiosis DOI 10.1007/s13199-014-0267-1

Responses: Thanks for your valuable comments and suggestions. In this study, tetracycline solution was brushed on tea buds and leaves with a thin film and it may osmosis into the plant tissue of stain in the surface. So that the tea green leafhoppers must take in tetracycline when they were feed. It is more conducive to work on the elimination of the symbiotic bacteria wolbachia in the insect body. In fact, we did initially try the way you mentioned by immersing the tea branches root(bottom) in a tetracycline solution. However, the function of sterilization on Wolbachia was not ideal even at a high concentration. It takes10mg/mL(10 times of the present smear method) and takes longer to eliminate the bacteria. The possible reason we speculate might be tetracycline can be inhaled by plants but cannot be delivered. The result is as shown in the following graph where “aspiration method”indicating the method of immersing the roots tea branches in a tetracycline solution.

Point 15: L148 “using”

Responses: Done.

Point 16: L188 This phylogenetic tree has not information (bootstrap) to support the relation between strains. Authors should modify it and explain all the settings used in Mega in M&M section also.

Responses: Thanks for your valuable comments and suggestions. we rebuilt the evolutionary tree to describe the relationship between strains. The re-established evolutionary tree is shown below.

Point 17: L196 I think a single negative test could not assess the absence of a bacteria. Authors stated the proble in amplifying wsp gene in discussion section (line 349). Maybe the amplification of other genes could support these results (16S, FtsZ).

Responses: Thank you for your valuable comments and suggestions. As mentioned earlier, we tested the 16S, FtsZ, and wsp genes in the early stage. Neither 16S nor FtsZ detected bands, but only the wsp gene detected corresponding band.

Point 18: L197 Difference in the length of sequence is usually normal, but wsp is a coding gene, hence the aminoacid sequence need to be checked to exclude stop codons (not clear in M&M section). Furthermore obtained sequences have to be published in a genetic database (GenBank) to be available to other researchers. In addition, authors should declare what differences were found among the obtained sequences. That is a crucial aspect to explain the coiches made for crossing experiment.

Responses: Thank you for your valuable comments and suggestions. Through sequencing, we found that the sequence sizes of the samples in Meitan and Jiu'an were exactly the same, while the samples collected in Fenggang had different molecular fragment lengths and sequence bases from both. At present, we have not published it in the gene bank, and plan to upload it to other researchers for reference. Thank you for your guidance and ideas.

The differences among the obtained sequences were listed in the graph in the revised version.

Point 19: L204 Please use the italics in the right way for species name and for wsp. Check the whole manuscript.

Responses: Thanks for your valuable comments and suggestions. The full text of species names and proper nouns such as wsp have been changed to italics.

 Point 20: L258 %infection decreased in a long time (10 days and 20 for 1 mg/mL and 10mg/ml respectively). Did the authors think about these results? Maybe the methodology is not appropriate due to the feeding strategy of leafhoppers?

Responses: Thanks for your valuable comments and suggestions. This experimental method has been modified on the basis. First, wolbachia in the tea green leafhopper is gradually eliminated as it continues to consume tetracycline solution. This experiment was carried out under the same environment and at the same time. A control group was designed during the experiment, and the concentration and dose effects were also relatively good. Obviously, it is therefore ruled out that the rearing strategy caused the difference in results.

 Point 21: L273-283 Results of paragraph should be taken in consideration in the whole work carried out. The positive differences found in the longevity (longer lifespan) of uninfected and natural-infected leafhoppers could be explained by the presence of other bacteria influencing the fitness of the host. the infection exclusion is an interesting aspect have to be taken in consideration. Other bacteria could be limited by the presence of wolbachia. the artificial absence of wolbachia (by tetracycline) could have give the opportunity to other bacteria to increase their population and so their effects on the host.

Responses: Thanks for your valuable comments and suggestions. This interesting idea was not considered before. It is indeed possible that other bacteria may be restricted by Wolbachia. Elimination of Wolbachia may increase the number of other bacteria and changed their impact on the host.

However, this result was still related to the cure of Wolbachia infection.

Point 22: L298 what does mean “worm”?

Responses: Thanks for your valuable comments and suggestions. We have revised the “worm” changed to “insect”.

 Point 23: L301 results in this paragraph are very ardous to understand. I would suggest to use a summary table to explain the crossings and use also different colors to indicate the strains.

Responses: Thanks for your valuable comments and suggestions. According to your suggestion, to visually see the difference between different combinations, I amended Fig. 9 to a table. Thank you for your guidance.

Table 1 The influence of different combination types on the number of offspring and sex ratio

Cross combination

Number of offspring

sex-ratio (female♀: male♂)

*W-♀×W

(14±0.25)d

1.67:1

W♀×W-

(20±0.32)b

1.59 :1

W-♀×W-

(25±0.46)a

1.22 :1

W♀×W

(18±0.28)c

1.50 :1

Note: Letters in the table indicate the significant difference when P≤0.05, and the data indicate the mean ± standard error. *Mating of infected male and uninfected females results in CI. ‘W’ represented the natural population of tea green leafhoppers. ‘W-’ represented the symbiosis removing populations.

Reviewer 3 Report

This manuscript reported the infection status of Wolbachia in the tea green hopper, Empoasca onukii using the PCR technique. The authors also investigated a geographic variation of the infection status and local existence of Wolbachia in an individual of Empoasca onukii. They also try to eliminate Wolbachia from Empoasca onukii by the tetracyclines treatment and reported the effect of Wolbachia on lifespan and reproduction of Empoasca onukii. However, the manuscript is unpolished, and I cannot agree with the conclusion of this manuscript.

Major points

Because any statistic was not shown throughout the manuscript, I cannot judge the validity of the data shown. Although the authors described that Duncan’s method was used, the p-value or any other statistic values were not shown. Moreover, Duncan’s method was often problematic. Similarly, what is the meaning of error bars in each figure? Is it SD, SE, or confidence interval?

The authors showed a phylogenetic tree in Fig. 1B. However, the bootstrap values were not shown. This prevents readers to judge the validity of this tree.

L335 The authors used the term “complete CI” to interpret the data on the tetracycline treatment. However, if the strain of Wolbachia causes the complete CI in Empoasca onukii, the number of offspring should be 0 in the combination of W-♀ and W♂. Why did the authors conclude that this strain of Wolbachia cause complete CI? The hatching rate should be investigated.

For the data of Fig. 9, the authors state that the strain of Wolbachia causes feminization of Empoasca onukii. However, I cannot understand why the authors exclude the possibility of male-killing in this data set.

Minor points

L29 Species name should be written by Italic.

L32 The term “hemimetabolous insects” is more common than “incompletely metamorphic insect”.

L47 What are mRNA viruses?

L55 Why does change in the male-to-female ratio result in a short insect life?

L61 What is “trm eat”?

L63 There is additional space in the sentence.

L63-65 I cannot understand the sentence completely. The number of black pupae was different between what and what? What is the black pupa?

L119 Please confirm the spell of H2O.

L155 What is “d”? Does it mean “day”?

L159 I think that the term “de-symbiotic” is not the usual term.

L193- The term “line” is a misspelled word of “lane”?

L193 This comma should be written by one-byte characters.

L206 RT-PCR is not real-time PCR.

L230 What is the chest? Does it mean thorax?

L246 Did Wolbachia exist in the abdomen of Empoasca onukii?

L276 The lifespan elongated of adults by the tetracyclines treatment might cause the elimination of harmful bacteria other than Wolbachia. Thus, the sentence “the symbiotic bacterium Wolbachia has a great impact on the lifespan of the adult” is an overstatement.

L289 The font size should be confirmed.

Fig. 9 The font of labels should be confirmed.

Fig. 9 What is “head” in the label of the left figure? It may be a translated unit meaning the number of individuals in Chinese, but this unit is not necessary for English.

L345 Although the infection rates of Wolbachia among populations were discussed, the data seemed not to be shown in results.

Title and Abstract: The scientific name of Empoasca onukii should be included.

Author Response

Responses to the Reviewer 3

Thanks for your kind dedicated work. All your suggestions have been carefully taken care of. And the English writing was polished and increased.

 Major points

Point 1: Because any statistic was not shown throughout the manuscript, I cannot judge the validity of the data shown. Although the authors described that Duncan’s method was used, the p-value or any other statistic values were not shown. Moreover, Duncan’s method was often problematic. Similarly, what is the meaning of error bars in each figure? Is it SD, SE, or confidence interval?

Responses: Thanks for your valuable comments and suggestions. The manuscript has been revised according to your suggestion. This part was revised as below: the experimental data are all statistical time and quantity statistics; the data is true and valid. The error bar represents the average ± standard error. In addition, the error bar represents the difference at P≤0.05. In addition, we have commented below the table or figure.

Point 2: The authors showed a phylogenetic tree in Fig. 1B. However, the bootstrap values were not shown. This prevents readers to judge the validity of this tree.

Responses: Good suggestions. We applied those related data and rebuilt the evolutionary tree and describe the relationship between strains with higher validity. The re-established evolutionary tree is shown below.

 Point 3: L335 The authors used the term “complete CI” to interpret the data on the tetracycline treatment. However, if the strain of Wolbachia causes the complete CI in Empoasca onukii, the number of offspring should be 0 in the combination of W-♀ and W♂. Why did the authors conclude that this strain of Wolbachia cause complete CI? The hatching rate should be investigated.

Responses: Thanks for your valuable comments and suggestions. As you questioned, we could not reach that conclusion and should delete the description about complete CI.  In addition, we added followingly” Wolbachia were not detected in the samples from Pingtang. Most likely causes might include the difference of geographical location, high altitude, climate change, also higher ultraviolet intensity.”

 Point 4 For the data of Fig. 9, the authors state that the strain of Wolbachia causes feminization of Empoasca onukii. However, I cannot understand why the authors exclude the possibility of male-killing in this data set.

Responses: Thanks for your valuable comments and suggestions.

Sure the feminization phenomenon might be indication of male-killing.   In addition, we have changed Fig. 9 to table 1 and make it more clear in description

Table 1 The influence of different combination types on the number of offspring and sex ratio

Cross combination

Number of offspring

sex-ratio (female♀: male♂)

*W-♀×W

(14±0.25)d

1.67:1

W♀×W-

(20±0.32)b

1.59 :1

W-♀×W-

(25±0.46)a

1.22 :1

W♀×W

(18±0.28)c

1.50 :1

Note: Letters in the table indicate the significant difference when P≤0.05, and the data indicate the mean ± standard error. ‘W’ represented the natural population of tea green leafhoppers. ‘W-’ represented the symbiosis removing populations.

Minor points

Point 1. L29 Species name should be written by Italic.

Responses: Thanks. We have revised the full text according to your requirements.

Point 2. L32 The term “hemimetabolous insects” is more common than “incompletely metamorphic insect”.

Responses: Thanks for your valuable comments and suggestions. We changed “incompletely metamorphic insect” to “hemimetabolous insects”.

Point 3. L47 What are mRNA viruses?

Responses: Sorry for the typo. It should be “RNA viruses”.

Point 4. L55 Why does change in the male-to-female ratio result in a short insect life?

Responses: Thanks for your valuable comments and suggestions.

The original expression here is that Wolbachia infection may affect the sex ratio in the offspring or may cause shorten of insect life span. So we revised the sentence as  “Some Wolbachia strains have negative effect on host by changing the sex ratio or resulting in a shorter insect life.”

Point 5. L61 What is “trm eat”?

Responses: sorry for the typo and “trm eat” was changed to “treat”.

 Point 6. L63 There is additional space in the sentence.

Responses: Thanks. We checked all existence of double space and have completed the revision as required, thanks!

 Point 7. L63-65 I cannot understand the sentence completely. The number of black pupae was different between what and what? What is the black pupa?

Responses: Thanks for your question and sorry for the obscure description. Black pupae is one of insect stage of Encarsia Formosa after its parasitize in the Bemisia tabaci. The offspring of Encarsia Formosa will get less when it was treated with tetracycline.

The original text was revised to " Stouthamer and Malk [34] found the offspring of tobacco whitefly (Bemisia tabaci) like their nymphs and pupae number decreased. Similarly, Zhou [35] also reported that when whitefly parasite (Encarsia formosa) was treated by tetracycline hydrochloride, its offspring would less. "

Point 8. L119 Please confirm the spell of H2O.

Responses: Done.

 Point 9. L155 What is “d”? Does it mean “day”?

Responses: Yes, it means "day ". we have revise it, thanks!

Point 10. L159 I think that the term “de-symbiotic” is not the usual term.

Responses:  Thanks. We agree. So we modified the description “through tetracycline de-symbiotic treatment” in line 407 to “through antibiotic treatment with tetracycline…” and “de-symbiotic” in line 19 was changed to “symbiosis removing”.

 Point 11. L193 The term “line” is a misspelled word of “lane”?

Responses: Thanks. Yes. The term “line” is a misspelled word of “lane”. All wrong ones have been modified as ‘lane’.

Point 12. L193 This comma should be written by one-byte characters.

Responses: Thanks. It has been modified according to your suggestion. And all related were modified.

 Point 13. L206 RT-PCR is not real-time PCR.

Responses: Thanks for your valuable comments and suggestions. All “RT-PCR” were changed to “real-time PCR”.

 Point 14. L230 What is the chest? Does it mean thorax?

Responses: Yes. It means insect thorax.

 Point 15. L246 Did Wolbachia exist in the abdomen of Empoasca onukii?

Responses: Yes. Wolbachia exist in the abdomen of Empoasca onukii as it can be seen in graph 5&6 in Fig. 4. Wolbachia can be detected in Empoasca onukii and it is mainly emerged in the abdomen part in adult stage as shown in fig. 3(B).

 Point 16. L276 The lifespan elongated of adults by the tetracyclines treatment might cause the elimination of harmful bacteria other than Wolbachia. Thus, the sentence “the symbiotic bacterium Wolbachia has a great impact on the lifespan of the adult” is an overstatement.

Responses: Thanks for your valuable comments and suggestions. We revised it as “The symbiotic bacterium Wolbachia has no significant effect on the egg stage and nymph stage of the tea green leafhopper and a slight impact on the lifespan of the adult when comparing to the natural population”.

 Point 17. Fig. 9 The font size should be confirmed.

Responses: Thanks, and sure we confirm the font size and adjusted them.

 Point 18. Fig. 9 The font of labels should be confirmed. Fig. 9 What is “head” in the label of the left figure? It may be a translated unit meaning the number of individuals in Chinese, but this unit is not necessary for English.

Responses: Thanks for your valuable comments and suggestions. With careful analysis on the data, we delete fig. 9 and changed the description form into table 1. And the differences between different combinations can be seen more clearly.

Table 1 The influence of different combination types on the number of offspring and sex ratio

Cross combination

Number of offspring

sex-ratio (female♀: male♂)

W-♀×W

(14±0.25)d

1.67:1

W♀×W-

(20±0.32)b

1.59 :1

W-♀×W-

(25±0.46)a

1.22 :1

W♀×W

(18±0.28)c

1.50 :1

Note: Letters in the table indicate the significant difference when P≤0.05, and the data indicate the mean ± standard error. ‘W’ represented the natural population of tea green leafhoppers. ‘W-’ represented the symbiosis removing populations.

Point 19. L345 Although the infection rates of Wolbachia among populations were discussed, the data seemed not to be shown in results.

Responses: Thanks for your valuable comments and suggestions. We added the infection rate description in the conclusion.

Point 20.  Title ang Abstract: The scientific name of Enpoasca onukii should be include.

Responses: Thanks, we have modified.

Round 2

Reviewer 1 Report

need extensive linguistic review

Author Response

No

Reviewer 2 Report

Dear Authors, I read the revised manuscript and, although it was improved, I think that the most important questions or suggestions I proposed were not answered or answered with a shallow attention.
I'll report my question your answer and my response about it (R2).

Point 5: L64 Please cite some papers where wolbachia was found in E. onukii. Is this the first 
attempt? If yes, authors should have excluded other symbiotic bacteria.

Responses: Thanks for your valuable comments and suggestions. In other leafhoppers have been 
reported in the presence of wolbachia, but the specific mechanism of action is rarely reported. 
This is the first report on wolbachia detection in tea green leafhoppers.The following is our list 
of literatures on leafhoppers.

R2: Why did the authors not state this important information? And, what about the other symbionts, 
especially bacteria? My previous suggestion was to check the presence of other bacteria (and, as you stated in 
the Ms, other symbionts such as microsporidia, viruses etc...). I think this is crucial especially because 
the results obtained show "a better fitness". A symbiont should help its host, otherwise it could be a pathogen or parasite.

Point 8: L93 5 adults= 1 sample? How the authors understand if the adults are all infected?

Responses: Thanks for your valuable comments and suggestions. Yes, we select the collected samples 
by selecting different geographical locations. During the experiment, one head of adults could not 
be detected with low infection indicator. So we enlarged the number of individual into 5 heads. 
We checked the general Wolbachia infection without the exact infection rate.

R2: based on FISH results, bacteria are present in the body, especially in chest and abdomen (please see Fig. 4 part 4 and 5).
If the use of a head was not useful in the dignosis of Wolbachia, why did you not try whole samples (with chest, abdomen)?

Point 9: L99 it is not clear why the authors studied only wsp gene. In “Multilocus sequence typing 
system for the endosymbiont Wolbachia pipientis” by Baldo et al, 2006, in Appl. Environ. Microbiol. 
Authors suggested the use of different genes to identify Wolbachia strains. In my opinion this is a 
crucial methodology to be applied in this kind of studies.

Responses: Thanks for your valuable comments and suggestions. In the early stage, we tested 16S, 
FtsZ, and wsp genes as shown in the following graph (date was not included in the research). Neither
16S nor FtsZ detected bands. Only the wsp gene detected the corresponding bands. As shown below:

Point 17: L196 I think a single negative test could not assess the absence of a bacteria. Authors 
stated the proble in amplifying wsp gene in discussion section (line 349). Maybe the amplification 
of other genes could support these results (16S, FtsZ).

Responses: Thank you for your valuable comments and suggestions. As mentioned earlier, we tested the
16S, FtsZ, and wsp genes in the early stage. Neither 16S nor FtsZ detected bands, but only the wsp 
gene detected corresponding band

R2: I cannot understand your methodology. It is the first report of wolbachia in tea green leafhoppers.
16S gene is the most amplifiable in bacteria detection. If you have problems in its amplification is 
probably due to the presence of other bacteria in the samples. And FTSZ could be amplified with 2 (see Baldo 2006) or 3 pairs of primers. 
Furthermore, a single negative test could only assess the absence of WOlbachia (if you use only wsp) and, if FTSZ and 16S did not give any 
result in all the samples, they cannot be considerate as "negative" in an unifected population. 

Point 16: L188 This phylogenetic tree has not information (bootstrap) to support the relation between 
strains. Authors should modify it and explain all the settings used in Mega in M&M section also.

Responses: Thanks for your valuable comments and suggestions. we rebuilt the evolutionary tree to 
describe the relationship between strains. The re-established evolutionary tree is shown below.

R2: In my previous revision I suggested to be very precise in explaining the methods of phylogenetic 
analysis performed, Please refer to my first revision. The methods don't report: translation in aminoacid, selection of outgroup 
(absent in this analysis), selection of the other strains of Wolbachia (why did you choose these strains?), nucleotide (aminoacid) 
substitution model, bootstrap and go on. The phylogenetic analysis here reported (both in M&M and results) is very inadequate, I'll suggest to remove it.

Point 18: L197 Difference in the length of sequence is usually normal, but wsp is a coding gene, hence the aminoacid sequence need to be checked to 
exclude stop codons (not clear in M&M section). Furthermore obtained sequences have to be published in a genetic database (GenBank) to be 
available to other researchers. In addition, authors should declare what differences were found among the obtained sequences. 
That is a crucial aspect to explain the coiches made for crossing experiment.

Responses: Thank you for your valuable comments and suggestions. Through sequencing, we found that the sequence sizes of the samples in Meitan and Jiu'an were exactly the same, while the samples collected in Fenggang had different molecular fragment lengths and sequence bases from both. At present, we have not published it in the gene bank, and plan to upload it to other researchers for reference. Thank you for your guidance and ideas.

R2: Obtained sequences of coding gene have to be checked for stop-codons or pseudogenes. If you use sequences in a Ms, they  have to be submitted in a genic databases. 
This is one of the most important ethical point of a researcher: share results so that other researchers would use them.

Other minor suggestion are reported in the attached PDF.

Based on insufficient revision of the manuscript, I think authors should repeat most of the tests.

Hence I confirm that Ms needs a scrupulous and accurate Major revision.

Author Response

Dear reviewer, we are so appreciated your scrupulous review and great advices. We have taken care of all the revisions accordingly. And the English writing is promoted carefully.

Point 1 Round 1 Point 5: L64 Please cite some papers where Wolbachia was found in E. onukii. Is this the first attempt? If yes, authors should have excluded other symbiotic bacteria.

R2: Why did the authors not state this important information? And, what about the other symbionts, especially bacteria? My previous suggestion was to check the presence of other bacteria (and, as you stated in the Ms, other symbionts such as microsporidia, viruses etc...). I think this is crucial especially because the results obtained show "a better fitness". A symbiont should help its host, otherwise it could be a pathogen or parasite.

Responses: Thanks. We cite two papers about the Wolbachia infections in leafhopper in rice plant [12,13]and tea leafhopper in tea garden [40]:

However, information regarding Wolbachia infection in tea leaf hopper from tea agroecosystems in China has almost never been investigated. Only the previous survey of Mao et al. [40] about symbiotic bacterial infection in tea garden insects has been published so far. Here, we present data of Wolbachia infection status on tea green leafhoppers from different geo-graphical tea garden.

----[12] Wiwatanaratanabutr, I. (2015). Wolbachia infection in leafhoppers and planthoppers: Diversity, density and geographic distribution in tropical rice agroecosystems. Journal of Asia-Pacific Entomology, 18(2), 277–282.

---[13] Wangkeeree, J., Suwanchaisri, K., Roddee, J., & Hanboonsong, Y. (2020). Effect of Wolbachia infection states on the life history and reproductive traits of the leafhopper Yamatotettix flavovittatus Matsumura. Journal of Invertebrate Pathology, 177, 107490.

---[40] Mao YX, Tan RR, Wang YP, Chen X, Wang HJ, Huang DJ, Gong ZM. Analysis of the bacterial diversity in adults of Empoasca onukii (Matsumura) based on 16S rRNA sequences. Plant Protection, 2018, 44(3): 17-23.

We use the reported Wolbachia specific gene wsp (Wolbachia surface protein) to check the existence of Wolbachia. We will incorporate these factors into our study to further investigate the effects of different strains.

Point 2 Round 1 Point 8: L93 5 adults= 1 sample? How the authors understand if the adults are all infected?

R2: based on FISH results, bacteria are present in the body, especially in chest and abdomen (please see Fig. 4 part 4 and 5).If the use of a head was not useful in the dignosis of Wolbachia, why did you not try whole samples (with chest, abdomen)?

Responses: Thanks for your valuable comments and suggestions. We sure used the whole sample by grinding them homogeneously in liquid N2 as described in section 2.2.1. Five adult insects were included since one adult insect could not be detected. Maybe the expression of “head” was confusing. Besides we use this as body part “head”, also as a classifier word. The word “head” in last response (“one head of” or “5 heads”) is classifier. Sorry for the confusing expression. We had revised all the related expression.

Point 3 Round 1 Point 9: L99 it is not clear why the authors studied only wsp gene. In “Multilocus sequence typing system for the endosymbiont Wolbachia pipientis” by Baldo et al, 2006, in Appl. Environ. Microbiol. Authors suggested the use of different genes to identify Wolbachia strains. In my opinion this is a crucial methodology to be applied in this kind of studies.

Point 17: L196 I think a single negative test could not assess the absence of a bacteria. Authors stated the proble in amplifying wsp gene in discussion section (line 349). Maybe the amplification of other genes could support these results (16S, FtsZ).

R2: I cannot understand your methodology. It is the first report of wolbachia in tea green leafhoppers. 16S gene is the most amplifiable in bacteria detection. If you have problems in its amplification is probably due to the presence of other bacteria in the samples. And Ftsz could be amplified with 2 (see Baldo 2006) or 3 pairs of primers. Furthermore, a single negative test could only assess the absence of WOlbachia (if you use only wsp) and, if Ftsz and 16S did not give any result in all the samples, they cannot be considerate as "negative" in an unifected population.

Responses: As we replied in last round 16S rRNA and FtsZ were also included in the detection of infection. Science the both results were negative, we only adopted the Wsp for the detection. To make it clear we provided methods and result of applying 16S rRNA and FtsZ gene:

The wsp1 (81F: TGGTCCAATAAGTGATGAAGAAAC; 691R: AAAAATTAAACGCTACTCCA) 、16S rRNA (F: CGGGGGAAAATTTATTGCT;R:AGCTGTAATACAGAAAGGAAATCGCCA) And FtsZ (F: ATYATGGARCATATAAARGATAG; R: TCRAGYAATGGATTRGATA) [42,43] were used for amplification. The PCR reaction mixture contained: 1.0 μL of genomic DNA template, 5 μL of 10x buffer, 4 μL of dNTP (2.5 mM), 2.0 μL of each primer (10 nM of Primer F and Primer R), 0.5 μL of Pyrobest DNA Polymerase (5 U/μL), and ddH2O added to a final reaction volume of 50 μL. The PCR program was initially started by 32 cycles of amplification at 94 °C for 30 s, 55 °C for 30 s and 72 °C for 60 s; subsequently followed by one cycle of amplification at 94 °C for 30 s, 52°C for 30 s and 72 °C for 60 s; and a final extension at 72 °C for 60 min. PCR was performed using a 2720 Thermal Cycler (Applied Biosystems, USA). The PCR products were separated on a 1% (w/v) agarose gel, stained with ethidium bromide (EB) and photographed by a gel imaging system.

we tested 16S rRNA, FtsZ, and wsp genes as shown in the following graph (date was not included in the research). Neither 16S rRNA nor FtsZ detected bands. Only the wsp gene detected the corresponding bands. As shown below:

Fig. 1 The amplification based on the wsp gene of Wolbachia. The numbers 1, 2, 3, and 4 denote the samples collected in Fenggang, Jiu’an, Meitan, and Pingtang;

Fig. 2 The amplification based on the 16S rRNA and Ftsz gene of Wolbachia.

Point 4 Round 1 Point 16: L188 This phylogenetic tree has not information (bootstrap) to support the relation between strains. Authors should modify it and explain all the settings used in Mega in M&M section also.

R2: In my previous revision I suggested to be very precise in explaining the methods of phylogenetic analysis performed, Please refer to my first revision. The methods don't report: translation in aminoacid, selection of outgroup (absent in this analysis), selection of the other strains of Wolbachia (why did you choose these strains?), nucleotide (aminoacid) substitution model, bootstrap and go on. The phylogenetic analysis here reported (both in M&M and results) is very inadequate, I'll suggest to remove it.

Responses: Thanks for your valuable comments and suggestions. We understand there are many groups of Wolbachia classification. But we just detected the group–specific Wsp PCR primers without classification. And we have deleted the phylogenetic tree part of the article based on your comment.

Point 5 Round 1 Point 18: L197 Difference in the length of sequence is usually normal, but wsp is a coding gene, hence the aminoacid sequence need to be checked to exclude stop codons (not clear in M&M section). Furthermore obtained sequences have to be published in a genetic database (GenBank) to be available to other researchers. In addition, authors should declare what differences were found among the obtained sequences. That is a crucial aspect to explain the coiches made for crossing experiment.

R2: Obtained sequences of coding gene have to be checked for stop-codons or pseudogenes. If you use sequences in a Ms, they have to be submitted in a genic database. This is one of the most important ethical point of a researcher: share results so that other researchers would use them.

Responses: Thank you for your valuable comments and suggestions. The samples used in our experiment were from the same location during the same period, so the Wolbachia strain in this process is also the same. We will further determine the strain of this strain in the next study. We have modified this part of the content according to the evolutionary tree part.

Point 6 Line 2 I would suggest to avoid the parenthesis

Responses: Done.

Point 7 Line 12 I would suggest to avoid the parenthesis

Responses: Done.

Point 8 Line 19 usually the right terminology used most of paper related to symbionts is "cured population", "cured individuals/females/..."

Responses: Done. “Symbiosis-removing population” was changed to “cured population”.

Point 9 Line 32 "It is an hemimetobolous insect..."   But, is this information really crucial?

Responses: Thanks for your comments and suggestions. Usually, "the hemimetabolous insects were more harmful than holometabola insect to crops. That’s why mention this information here. Thanks!

Point 10 Line 46 Endosymbiosis could include bacteria...

Responses: Thanks. The sentence has been revised as “Some examples of endosymbiosis are bacteria”.

Point 11 Line 57 please check

Responses: Done.

Point 12 Line 59 is this right? If a male is uninfected, does it has a Wolbachia belonging to a strain? Please rephrase taking into account the cited references.

Responses: Thanks. The sentence has been revised as “CI occurs when a Wolbachia infected male mates with an uninfected female or between males and females infected with different Wolbachia strains, result in progeny that die during very early embryogenesis and reduce the number of insects.”

Point 13 Line 63 parasitoid

Responses: Done. “Parasite” was changed to “parasitoid”.

Point 14 Line 63 please add: authority (order: family)

Responses: Done. Encarsia formosa Gahan.

Point 15 Line 66 please add: authority (order: family)

Responses: Done. Bemisia tabaci Gennadius.

Point 16 Line 67 parasitoid

Responses: Done. “Parasite” was changed to “parasitoid”.

Point 17 Line 74 I think this sentence is a repetition of the previous one

Responses: Thanks for your comments and suggestions, we have removed it.

Point 18 Line 75 Delete “in”.

Responses: Done.

Point 19 Line 85 I think this sentence as a dot "." not useful

Responses: Done.

Point 20 Line 88 please check the English lenguage of this sentence

Responses: Done. “And the tea branches bottom was bathed in a nutrient solution [39] and the solution was replaced every 5 days to ensure the freshness of tea braches” was changed to “And the tea branches were cultured with nutrient solution, with the nutrient solution [41] changed every 5 days used to ensure the freshness of the tea branches.”.

Point 21 Line 96-110 Please check the English used in these sentences. I think there is an abuse of "then", "add" "added". I would suggest an extenssive editing of English language and style.

Responses: we have rewritten this paragraph.

Point 22 Line 96 “f”

Responses: Done.

Point 23 Line 97 Delete “the bottom of”.

Responses: Done.

Point 24 Line 97 ground (without it)

Responses: Done.

Point 25 Line 97 proteinase

Responses: Done. “Protenase” was changed to “proteinase”.

Point 26 Line 99 Delete “the tube”

Responses: Done.

Point 27 Line 114 please check the reference cited for the primers

Responses: Thanks, we have modified to “Zhou, W.; Rousset, F.; O'Neill, S. Phylogeny and PCR-based classification of Wolbachia strains using wsp gene sequences. Proc. R. Soc. Lond. B 1998, 265: 509-515.”

Point 28 Line 139 “used”

Responses: Done. “applied” was changed to “used”.

Point 29 Line 148 “and?”

Responses: Yes. Hydrogen peroxide(H2O2) was solved in the ethanol to make 6% solution in volume.

Point 30 Line 150 “?”

Responses: We revised it as “…absolute ethanol and PBST [(PBST: PBS with 0.2% Tween-20), ethanol: PBST=1:1(v/v)].

Point 31 Line 211-215 The phylogenetic tree gives different information respect to what stated by the authors. Meitan/Jiuan are not related to Sycoscapter moneres. They belong to another clade.

Responses: Yes. You are corrected. We had made a wrong station. According to the phylogenetic tree, Meitan/Jiuan are not related to Sycoscapter moneres. And they belong to another clade. In respect to your suggestion “this phylogenetic tree is of little significance”, we have removed it, thanks!

Reviewer 3 Report

The manuscript has been improved at several points. However, serious problems remain. Especially, the method and statistical analysis of the mating experiment should be explained in detail. Because the sample size often is unclear, I cannot evaluate the validity of the experiment.

Major points
L182 Did the authors confirm the infection status of all insects in the mating experiment by PCR or the other technique?

Fig.1B How did the authors collect the sequence data of other insects? The method should be written.

L206 How did the authors conclude that the insects collected in Pingtang were not infected by Wolbachia? What did assure that DNA extraction is succeeded?

L212 The authors showed a phylogenic tree based probably on the DNA sequence, but they seemed to use the amino acid sequence according to the main text. Which did the author use?

L214 The authors pointed out that the insects collected in Meitan and Jiu’an had similar alleles with Sycoscapter moneres, however, the samples from Meitan and Jiu’an were in a different clade with S. moneres in Fig.1B. The authors should explain this discrepancy.

On the analysis:
I cannot understand the statistics used in the manuscript. What is Duncan’s method? Is it Duncan’s multiple range test? If so, the authors should show p-values among groups analyzed using symbols or other methods. 

On the analysis:
Duncan’s multiple range test has a great risk of type I error. Do the authors recognize this problem?

On the analysis:
The authors showed the standard errors when the p-value was <0.05. However, the standard errors always should be shown.

Fig.3c Although the author mention only Duncan’s method in the materials & methods, the method was not adequate in Fig. 3c because this dataset includes only 2 groups. The authors should confirm the analysis throughout the manuscript.

Fig. 7a The authors showed the standard errors, although they pointed out that there is no significant difference in the main text. This inconsistency confuses the readers.

L330 The number of offspring in the W♀- and W♂ group differed from Table 1.

L328 Because the number of offspring was not defined sufficiently in this manuscript, I cannot agree with the conclusion that the obvious CI is observed. How did the authors count the number of offspring? The difference in the number of offspring could be generated by the CI, male-killing, and the others mechanism.

L337 I did not understand the conclusion that the Wolbachia strains cause male-killing or feminization. Where is the statistical analysis? Duncan’s multiple range test cannot be applied to the analysis of sex ratio. Where is the p-value? Where is the sample size?

L365 The term “chest” is not a proper term meaning the thorax of insects.

Minor points
L12 The scientific name should not be enclosed between parentheses. 
L19 The term “symbiosis removing population” is not proper in English.
L21-22 The present data did not support this sentence.
L214, 215 A part of the genus name is not italic, and the first character of the genus name should be capitalized.

Author Response

Dear reviewer, we are so appreciated your scrupulous review and great advices. We have taken care of all the revisions accordingly. And the English writing is promoted carefully.

Major points

Point 1 Line 182 Did the authors confirm the infection status of all insects in the mating experiment by PCR or the other technique?

Responses: Yes. We detected all the insects by PCR after mating and dying. If cured population (that should be no infection) were found with Wolbachia infection, The data of this group will be removed. And all the cured population were 100% uninfected.

Point 2 Fig.1B How did the authors collect the sequence data of other insects? The method should be written.

Responses:  Sequence data were acquired by the analysis tool BLAST in NCBI. And all the most related insects’ sequences were listed in this phylogenetic tree to compare. And In respect to another reviewer’s suggestion “this phylogenetic tree is of little significance”, we removed it in this revision, thanks!

Point 3 L206 How did the authors conclude that the insects collected in Pingtang were not infected by Wolbachia? What did assure that DNA extraction is succeeded?

Responses: During the detection experiment process, we repeatedly tested many times and verified the detection results especially in sample from Pingtang. All 16S rRNA, Ftsz and Wsp (Wolbachia surface protein) gene detection were negative.  Though it could be a false negative if the density infection is so low that could not be examined with our method. But compare with other samples at same condition it did was undetected. So we concluded that insects collected in Pingtang were not infected by Wolbachia.   

Point 4 L212 The authors showed a phylogenic tree based probably on the DNA sequence, but they seemed to use the amino acid sequence according to the main text. Which did the author use?

Responses: Yes. We used DNA sequence to contrast the phylogenetic tree. The description of "amino acid sequence" in the main text was wrong and should modified to "DNA sequence". But this descripting and phylogenetic tree was suggested to be removed since the phylogenetic tree was of little significance. Thank you for pointing out our problem.

Point 5 L214 The authors pointed out that the insects collected in Meitan and Jiu’an had similar alleles with Sycoscapter moneres, however, the samples from Meitan and Jiu’an were in a different clade with S. moneres in Fig.1B. The authors should explain this discrepancy.

Responses: Yes. We are corrected. We had made a wrong station. According to the phylogenetic tree, Meitan/Jiuan are not related to Sycoscapter moneres. And they belong to another clade. In respect of discrepancy between them, we did not check the reason why the Wolbachia in insects in different place are not same clade.

Point 6 On the analysis: I cannot understand the statistics used in the manuscript. What is Duncan’s method? Is it Duncan’s multiple range test? If so, the authors should show p-values among groups analyzed using symbols or other methods. 

Responses: We use Duncan's new multiple range test. Duncan's new multiple range test method is uses a series of the shortest significant difference Rp to compare the difference between two averages. Now, we changed to “used SPSS 19.0 software for statistical analysis of the experimental data, the data was subjected to a one-way analysis of variance (p<0.05), and Data were expressed as the mean ± standard error, followed by a significant difference test (LSD)”. In addition, I have marked the lowercase letters on the data map in the text and the corresponding revisions in the text.

Point 7 On the analysis: Duncan’s multiple range test has a great risk of type I error. Do the authors recognize this problem?

Responses: Thank you for your comments and suggestions. Duncan’s multiple range test has Type I and Type II errors, which cannot be avoided. We modify the data processing process to reduce the occurrence of errors, and revised to“data were expressed as the mean ± standard error, and the data were subjected to one-way analysis of variance (ANOVA)(p<0.05) followed by a significant difference test (LSD) using SPSS statistics 19.0”.

Point 8 On the analysis: The authors showed the standard errors when the p-value was <0.05. However, the standard errors always should be shown.

Responses: Thanks for your valuable comments and suggestions. We have used lowercase letters on all charts in the text to represent standard errors.

Point 9 Fig.3c Although the author mention only Duncan’s method in the materials & methods, the method was not adequate in Fig. 3c because this dataset includes only 2 groups. The authors should confirm the analysis throughout the manuscript. Fig. 7a The authors showed the standard errors, although they pointed out that there is no significant difference in the main text. This inconsistency confuses the readers.

Responses: Thanks for your valuable comments and suggestions. We have perfected.

Point 10 L330 The number of offspring in the W♀- and W♂ group differed from Table 1.

Responses: Thanks for your valuable comments and suggestions. This was a wrong typing and we have checked and revised.

Point 11 L328 Because the number of offspring was not defined sufficiently in this manuscript, I cannot agree with the conclusion that the obvious CI is observed. How did the authors count the number of offspring? The difference in the number of offspring could be generated by the CI, male-killing, and the others mechanism.

Responses: Yes, we agree with you that it could not be concluded that “obvious CI is observed.” If their original sample size were given. In fact, the “number of offers ring” given in the table were actually “the number of offspring from per mate”. This would be clear to observe CI wich caused lower number of offer spring. So, in the revised MS, we provided the number of mate groups in Table1 and the number of offspring could be then defined sufficiently. In this process,we collected and counted the total number of offspring and check the insect sex ratio under microscope(difference in the abdomen).

Point 12 L337 I did not understand the conclusion that the Wolbachia strains cause male-killing or feminization. Where is the statistical analysis? Duncan’s multiple range test cannot be applied to the analysis of sex ratio. Where is the p-value? Where is the sample size?

Responses: Thank you for your comments and suggestions. We appreciated your suggestion. You are right that we do not provide the sample numbers of the offspring and there was no statistical analysis. In this revision, we provided offspring numbers of the male and female with statistical analysis respectively in Table 1.

Mating type

No. of mates tested

No. of offspring of per mate

offspring of per mate

No.of Female (♀)

No. of Male (♂)

sex-ratio (female♀: male)

W-♀×W

17

(16±0.25)d

(10±0.05) b

(6±0.02) c

1.67:1

W♀×W-

21

(20±0.32)b

(12±0.06) ab

(8±0.03) b

1.50:1

W-♀×W-

16

(25±0.46)a

(14±0.07) a

(11±0.05) a

1.27:1

W♀×W

19

(18±0.28)c

(11±0.05) b

(7±0.03) b

1.57:1

Point 13 L365 The term “chest” is not a proper term meaning the thorax of insects.

Responses: Thanks. We agree with and we change all “chest” to be “thorax”.

Minor points

Point 14 L12 The scientific name should not be enclosed between parentheses. 

Responses: Thanks. We avoided using of parenthesis in scientific name.

Point 15 L19 The term “symbiosis removing population” is not proper in English.

Responses: Thanks. “Symbiosis removing population” was changed to “cured population”.

 Point 16 L21-22 The present data did not support this sentence.

Responses: According to our response to above question and with more data provided in table 1. This conclusion could be proposed. And we revised the “Genetic research results confirmed that…” to be “results from population reproduction research confirmed that…”

Point 17 L214, 215 A part of the genus name is not italic, and the first character of the genus name should be capitalized.

Responses: Thanks. All related editions were taken care of.
